# Genome-wide real-time in vivo transcriptional dynamics during *Plasmodium falciparum* blood-stage development

Heather J. Painter[1,2], Neo Christopher Chung [3,7], Aswathy Sebastian[4], Istvan Albert[1], John D. Storey[3,5] & Manuel Llinás[1,2,6]

Genome-wide analysis of transcription in the malaria parasite *Plasmodium falciparum* has revealed robust variation in steady-state mRNA abundance throughout the 48-h intraerythrocytic developmental cycle (IDC), suggesting that this process is highly dynamic and tightly regulated. Here, we utilize rapid 4-thiouracil (4-TU) incorporation via pyrimidine salvage to specifically label, capture, and quantify newly-synthesized RNA transcripts at every hour throughout the IDC. This high-resolution global analysis of the transcriptome captures the timing and rate of transcription for each newly synthesized mRNA in vivo, revealing active transcription throughout all IDC stages. Using a statistical model to predict the mRNA dynamics contributing to the total mRNA abundance at each timepoint, we find varying degrees of transcription and stabilization for each mRNA corresponding to developmental transitions. Finally, our results provide new insight into co-regulation of mRNAs throughout the IDC through regulatory DNA sequence motifs, thereby expanding our understanding of *P. falciparum* mRNA dynamics.

[1] Department of Biochemistry and Molecular Biology, The Pennsylvania State University, University Park, PA 16802, USA. [2] Huck Center for Malaria Research, The Pennsylvania State University, University Park, PA 16802, USA. [3] Lewis-Sigler Institute for Integrative Genomics and Department of Molecular Biology, Princeton University, Princeton, NJ 08544, USA. [4] Huck Institutes of the Life Sciences, The Pennsylvania State University, University Park, PA 16802, USA. [5] Center for Statistics and Machine Learning, Princeton University, Princeton, NJ 08544, USA. [6] Department of Chemistry, The Pennsylvania State University, University Park, PA 16802, USA. [7] Present address: Institute of Informatics, Faculty of Mathematics, Informatics, and Mechanics, University of Warsaw 02-097 Warsaw, Poland. Correspondence and requests for materials should be addressed to M.Lás. (email: manuel@psu.edu)

Hundreds of millions of annual malaria cases are caused by intracellular protozoan *Plasmodium* parasites resulting in almost half a million deaths and an enormous economic impact on almost half the world's population annually[1]. Efforts to eradicate this pathogen have been at the forefront of infectious disease research for decades. Although it has been over 15 years since the completion of the whole genome sequence[2] of *Plasmodium falciparum*, transcriptional regulation throughout the parasite's complex life cycle remains poorly understood.

The malaria parasite life cycle commences upon the transfer of *Plasmodium* sporozoites from the saliva of an infected *Anopheles* spp. female mosquito vector to a human host. These sporozoites travel to and colonize a small number of hepatocytes within the liver where they replicate, forming thousands of merozoites. Liver-stage merozoites are released into the circulatory system and initiate the blood stage of infection, which is responsible for the clinical symptoms of malaria. Development during the 48 h intraerythrocytic development cycle (IDC) is comprised of parasite maturation and cell division, resulting in the formation of up to 32 daughter cells, which are released and invade new erythrocytes[3]. A small proportion of asexual parasites commit to an alternative cellular fate of sexual differentiation during the IDC, eventually resulting in the development of male and female gametocytes. Maturation of *P. falciparum* sexual stage parasites takes 10–12 days, and it is these sexual forms that are taken up by a mosquito vector for eventual onward transmission to another human host, thus completing the *Plasmodium* full life cycle[4]. Differentiation of the malaria parasite in these various cell types and tissues is regulated by coordinated programs of gene expression[5–10].

Gene expression during the IDC is highly periodic and follows a transcriptional cascade with the majority of genes expressed in a "just-in-time" fashion[5,10,11]. Although canonical transcriptional machinery is present, the mechanisms that underlie the regulation and specificity of active transcription remain largely uncharacterized[12,13]. To date, only a single family of specific transcription factors, the 27-member Apicomplexan AP2 (ApiAP2) protein family, has emerged as transcriptional regulators with functions across all developmental stages[12,14]. However, the challenge of regulating roughly 5500 *Plasmodium* genes with a small repertoire of transcription regulators suggests a significant role for epigenetic control and post-transcriptional regulation in *Plasmodium* spp.[13,15–18]. The control of gene expression at the RNA level is evidenced by several studies reporting a significant delay between transcription and translation[19–22], ribosomal influence on the timing of mRNA translation[23,24], and a lack of coordination between active transcription and mRNA abundance during parasite development[25,26]. Bioinformatic analyses have suggested that between 4 and 10% of the *Plasmodium* genome encodes RNA-binding proteins (RBPs)[27,28]. While the majority of predicted RBPs remain uncharacterized, several post-transcriptional regulatory factors have been well-studied such as DOZI (DDX-6 class DEAD box RNA helicase), CITH (Sm-like factor homolog of CAR-I and Trailer Hitch), and pumilio family proteins (PUF1 and PUF2) that have been demonstrated to play critical roles in the translational repression of genes essential for *Plasmodium* transmission[29]. Additionally, the acetylation lowers binding affinity (ALBA) domain-containing proteins (*Pf*Alba1-4)[30–32], Bruno/CELF family members (CELF1 and CELF2)[33], and Pfg27[34] have been demonstrated to interact with RNA; however, the exact role has yet to be defined for a majority of these RBPs in parasite development.

In all living systems, cellular mRNA levels are determined by the interplay of tightly regulated processes of RNA production, processing, stabilization, and degradation[35]. Thus, changes in transcript abundance measured by DNA microarray, RNA-sequencing, or RT-qPCR studies reflect the relative rates of both transcription and mRNA stabilization. Since the advent of the complete *P. falciparum* genome sequence, numerous studies have utilized whole-genome methods to measure transcript abundance from population of parasite-infected red blood cells[9,10,19,36] as well as, more recently, single cells[37,38]. In addition, other studies have measured transcriptional activity[26,39], mRNA half-lives[40], RNA Pol II binding[41], and transcription start sites[42] throughout development. Although these studies have revealed many important details of the regulation of gene expression in *Plasmodium* during the IDC, none capture mRNA dynamics on a whole-genome scale without perturbations to the parasite. Measurement of mRNA half-lives is commonly performed following chemical inhibition of transcription, which induces a stress response or cellular death thereby impacting the ability to accurately measure physiological mRNA turnover parameters[43]. Similarly, whole genome capture of transcriptional activity via nuclear run-on[39] or GRO-seq[26] methods require isolation of nuclei or permeabilization of cells, which compromises cellular physiology and nascent nuclear transcriptional activity. Therefore, we characterized the mRNA dynamics of the *P. falciparum* transcriptome by comprehensively capturing nascent mRNA transcription, stabilization, and total abundance of mRNA in vivo throughout the IDC using 4-thiouracil (4-TU) biosynthetic RNA labeling.

Biosynthetic labeling of pyrimidines has revolutionized the capture and measurement of in vivo cellular mRNA dynamics during development and in response to various stressors[44–46] across a diverse array of organisms[45,47,48] including *Plasmodium*[25]. In human and yeast cells, short periods of metabolic labeling into nascent RNA has been used to capture RNA synthesis, decay, and splicing rates[45,49,50]. However, thiol-modified RNA capture and kinetic profiling of RNA dynamics are dependent upon pyrimidine salvage, and the evolutionary lineage of *Plasmodium* has lost this biochemical capacity[51]. We have previously demonstrated that genetic engineering of the pyrimidine salvage pathway into *Plasmodium* parasites enables the efficient uptake of 4-TU and biosynthetic capture of thiol-labeled RNAs to examine RNA dynamics during parasite development[25].

Herein, we report the application of biosynthetic RNA labeling and capture[25] to measure genome-wide nascent mRNA transcription and stability in vivo throughout the IDC of the malaria parasite. This method relies on the genetically facilitated scavenging of 4-TU and genome-wide DNA microarray measurements of labeled and unlabeled RNAs. Using this method, we have captured mRNA dynamics during asexual development at hourly resolution and apply a quantitative normalization model to define gene-specific mRNA metabolism. These data provide dynamic transcription and stability information for all mRNAs expressed during the IDC and also recapture known features of the "just-in-time" cascade of mRNA transcription. Additionally, modeling of the mRNA dynamics reveals regulatory mechanisms that cooperate during parasite development, which we used to predict functional roles for putative transcriptional regulators. Our results provide an updated in-depth view of mRNA transcriptional dynamics and gene regulation in *P. falciparum* throughout the IDC.

## Results

**Genetic modification of *P. falciparum* for pyrimidine salvage.** In vivo mRNA labeling is dependent upon the salvage of 4-TU; however, *P. falciparum* is not capable of pyrimidine precursor uptake and incorporation into RNA[2,51]. We previously reported the use of episomal expression of the bifunctional *f*usion gene encoding *c*ytosine deaminase and *u*racil phosphoribosyltransferase (FCU) from yeast to enable parasite salvage of 4-TU[25]. To avoid cell-to-cell heterogeneity in plasmid copy number expressing FCU, this study uses a single copy cassette that is stably integrated into the attB

locus of 3D7[attb52] to constitutively express a GFP-tagged FCU protein (Supplementary Fig. 1A)[25]. Stable integration of *fcu-gfp* (3D7[attB::fcu-gfp]) was verified by PCR (Supplementary Fig. 1A). Expression of FCU-GFP was confirmed by Western blot and localized to the cytoplasm by immunofluorescent imaging analysis (Supplementary Fig. 1B and C). Salvage and incorporation of a 4-TU into total cellular RNA by the 3D7[attB::fcu-gfp] transgenic parasites was confirmed by Northern blot (Supplementary Fig. 1D).

**Nascent in vivo mRNA labeling and capture throughout the IDC.** The goal of this study was to produce high-resolution RNA dynamics data for each transcript expressed during the 48 h *P. falciparum* IDC transcriptome. To capture nascent RNA transcription and stabilizat ion, we exposed highly synchronized 3D7[attB::fcu-gfp] to 10 min pulses of 40 μM 4-TU followed by total RNA extraction and capture of labeled and unlabeled transcripts (Fig. 1a and Supplementary Fig. 2A)[25]. We chose a short labeling time of 10 min based on the time required for 4-TU uptake and incorporation into 4-thiol-RNA transcripts[25]. When the length of 4-TU incubation is sufficiently short, the labeled RNA is likely still in the nucleus and subjected to minimal degradation, and thus is reflective of the average transcription rate[49]. To capture nascent transcription throughout the IDC, 10 min pulses were repeated on individual timepoints every hour for the full 48 h developmental cycle (Fig. 1a) followed by total RNA extractions for each hourly sample and biotinylation of the thiolated RNA transcripts. Nascently transcribed (4-TU-labeled, biotinylated) mRNA was affinity-captured via streptavidin magnetic beads (Supplementary Fig. 2A)[25,46]. For each timepoint, the genome-wide profiles for the resulting RNA population, total, labeled (transcribed), and unlabeled (stabilized) RNAs were quantified using DNA microarray analysis (Supplementary Fig. 2A). The RNAs present in the cell that were not modified during the 10 min 4-TU labeling pulse are designated as "stabilized" throughout

this study. However, we note that our method cannot distinguish whether these "stabilized" RNAs are free or bound by RBPs and are likely turning over at variable rates. Therefore, the term "stabilized" herein is simply used to describe the persistence of an unlabeled transcript that was present prior to the labeling pulse.

To investigate whether in vivo RNA labeling perturbed transcription dynamics, we correlated the total RNA abundance levels from the 4-TU treated 3D7[attB::fcu-gfp] timecourse with an earlier *P. falciparum* transcriptome from untreated wild-type 3D7 cells[10] and found no significant changes in the pattern of RNA levels during the IDC (median Pearson corr = 0.72, Supplementary Fig. 2B). This suggests that transcription or steady-state mRNA levels are not perturbed in the presence of a short pulse of 4-TU. We next determined if there was bias in our RNA labeling due to the brevity of the 4-TU pulse by comparing gene length and dUTP content to intensity of the signal captured (Supplementary Fig. 2C and S2D). Although the distribution of gene length and dUTPs/per gene distribution is broad (23-30864 bps and 7-9242 dUTPs/gene, respectively; Supplementary Fig. 2C), there is no significant bias in the mRNAs that are labeled with 4-TU and captured by biotin–streptavidin interaction (Supplementary Fig. 2D). Therefore, the chosen 10 min pulse length allows for the capture and accurate measurement of nascent transcription genome-wide without further normalization. Overall, genome-wide biosynthetic labeling and capture throughout the IDC revealed nascent transcription profiles for 5198 genes (91.4% of *P. falciparum* annotated transcripts).

**Statistical model of *P. falciparum* asexual mRNA dynamics.** Because standard two-color DNA microarray analysis normalizes $\log_2$(green/red) values within individual timecourses, the $\log_2$ values do not distinguish the relative contributions of the various pools of RNA (labeled, unlabeled) to the total RNA signal (Fig. 1b; raw $\log_2$ ratios) when timecourses are directly compared

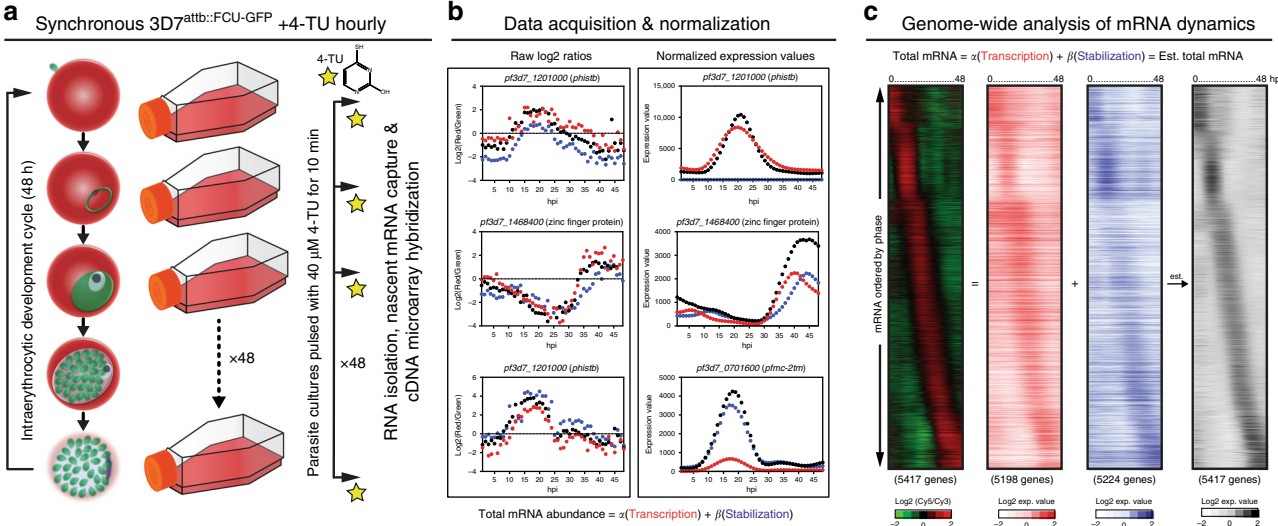

**Fig. 1** Biosynthetic mRNA labeling and capture throughout the *P. falciparum* IDC reveals complex developmental mRNA dynamics. **a** Biosynthetic mRNA capture timecourse with 4-TU (40 μM) labeling (10 min pulse) of highly synchronous transgenic 3D7[attB::fcu-gfp] parasites every hour throughout the 48 h IDC followed by total RNA extraction. 4-TU labeled RNA is separated from the total RNA resulting in two fractions, transcribed (labeled) and stabilized (unlabeled). These RNAs and the total cellular RNA pool were quantified by DNA microarray analysis (Supplementary Fig. 2). **b** Data obtained by cDNA microarray was normalized and modeled to determine the contribution of nascent transcription (red circles) and stabilization (blue circles) to the total abundance profile (black circles) for each gene. **c** The modeled expression values of nascent transcription (5207 genes, red) and stabilization (5236 genes, blue) values for each gene were ordered based on the peak timing of total abundance (5428 genes) during the 48 h IDC as determined by Lomb-Scargle analysis and displayed in a heatmap as the $\log_2$ expression value. The summation of the expression values for both transcription and stabilization is displayed as a heatmap representing the estimated total mRNA abundance (5207 genes, black), recapitulating the canonical phaseogram of measured total mRNA abundance (left, RGB heatmap)

to each other. To circumvent this issue, we developed a statistical model to quantitatively determine the contribution of nascent transcription (labeled) and mRNA stabilization (unlabeled) to the total abundance profile of each gene by comparing the signal intensities for each probe on the DNA microarrays across all three timecourses (Methods). After the three timecourses were separately normalized by Rnits[53] (Supplementary Fig. 3), the mRNA dynamics were estimated by non-negative least squares since the contribution of unlabeled and labeled mRNA to the total cellular RNA is always additive (Supplementary Fig. 3). The balance between nascent transcription and mRNA stabilization is estimated by optimizing the following equation:

$$\text{Total mRNA abundance} = \alpha(\text{nascent transcription})$$
$$+ \beta(\text{mRNA stabilization})$$

Estimated coefficients for $\alpha$ and $\beta$ were utilized to calculate the expression value of nascent transcription and mRNA stability, respectively (Supplementary Fig. 3). Expression values representing the total abundance, transcription, and stabilization were averaged across the gene to which they mapped using Tukey's biweight[54,55] and plotted across the 48 h IDC (Fig. 1c, calculated expression values), resulting in mRNA dynamic profiles for most *P. falciparum* genes (Supplementary Data 1). Gene plots can be found on individual gene pages on http://www.PlasmoDB.org[56] and https://doi.org/10.6084/m9.figshare.6200792.v1 where both the raw log$_2$(green/red) ratios and normalized expression values are also available for download.

Based on the calculated normalized nascent mRNA expression values, we found that active transcription from the nuclear genome occurs throughout the IDC (Fig. 1c and Supplementary Data 1). We also found that transcription of nascent mRNAs from both the mitochondrial and apicoplast organellar genomes is tightly coordinated and begins 8–10 hpi (Supplementary Fig. 4A and 4B). Transcription of these non-nuclear genes peaks at 31 hpi (Supplementary Fig. 4B), coinciding with the start of organellar genome replication and division[57,58]. In contrast, nuclear-encoded genes destined for the apicoplast are transcribed prior to apicoplast encoded genes (Supplementary Fig. 4C), likely due to their role in transcription and translation of the apicoplast genome[59]. Interestingly, we also observe that the capture of nascent transcription from the apicoplast genome reveals a wide range of expression levels regardless of their genome position (Supplementary Fig. 4D).

Based on the statistical model of the data, we found that the abundance profiles of individual genes are dictated by a combination of nascent transcription and mRNA stability as the parasite develops (Fig. 1b). When considering the global transcriptome, the calculated contribution (from the statistical model) of nascent transcription and mRNA stabilization to total abundance profiles varies as the parasite progresses through the IDC (Fig. 1c and Supplementary Fig. 5). To confirm the accuracy of the model, we summed the expression values of both nascent transcription and stabilization to simulate the total RNA values (Fig. 1c). When the estimated total mRNA abundances are plotted in the same order as the observed values, the canonical phaseogram of the *P. falciparum* transcriptome observed by traditional methods is recapitulated (Fig. 1c)[5,9,10,36], with a median correlation of 0.94 (Supplementary Fig. 5A). Additionally, the gene-specific patterns of mRNA stabilization or transcription correlate with the overall transcript abundance profiles (Supplementary Fig. 5B and 5C; transcription versus total abundance median corr = 0.86, stabilization median corr = 0.85), confirming that our method and model capture the mRNA dynamics that dictate the abundance of a given transcript throughout the IDC. In contrast, gene-specific transcription and stabilization dynamic

profiles are not highly correlated with each other (median $r = 0.69$) (Fig. 2a), likely due to the differences in peak timing of nascent transcription versus stabilization. By calculating the difference in the maximum transcription and stabilization to the total abundance, we estimate that peak transcription precedes the peak abundance (−80 min), while peak stabilization follows approximately 41 min after (Fig. 2b). This supports the current notion that the majority of genes are transcribed in a "just-in-time" manner with nascent transcription peaking prior to mRNA abundance, while post-transcriptional regulation likely influences the delay from peak time of mRNA abundance.

**Variation in temporal mRNA dynamics throughout the IDC.** To determine the stage-specific effects of mRNA dynamics on transcript abundance, we binned genes into six groups based on their peak timing of abundance during the IDC as captured by the total RNA timecourse (Fig. 2c). These six groups reflect the major morphological stages of development during the IDC including early ring (0–10 hpi), mid-ring (11–15 hpi), late ring/early trophozoite (16–21 hpi), mid-trophozoite (22–26 hpi), late trophozoite (27–32 hpi), and schizont (33–48 hpi). We then predicted the predominant ($r \geq 0.50$) mechanism acting on each gene based on the correlation of either transcription (Fig. 2c, red) or stabilization (blue) to the total abundance profile (Fig. 2c, pie charts). Overall, the influence of transcriptional activity or mRNA stability on total transcript abundance across the genome is relatively equal (transcription = 2767 genes; stabilization = 2660 genes) (Fig. 2c). However, there is a higher proportion of contribution from nascent transcription to mRNA abundance profiles at a majority of developmental stages (early ring = 55%, late ring/ early trophozoite = 68%, mid-trophozoites = 61%, late trophozoites = 52%), while stabilization dominates at the mid-ring (64%) and schizont (62%) stage transitions (Fig. 2c and Supplementary Fig. 5D). These results reveal that there are stage-specific regulatory mechanisms that influence the patterns of mRNA abundance through both transcription and mRNA stability during the IDC.

Previous studies have suggested that both transcriptional activity and mRNA half-life increase as the parasite progresses through the IDC[39,40]. These results were based on methodologies that relied on measuring ex vivo transcription from isolated nuclei[26,39] or inhibition of transcription with Actinomycin D[40] followed by RNA-seq or DNA microarray to quantify mRNA. Because our dataset was generated without physical or chemical perturbations to the parasite, we analyzed our results in the context of these earlier studies[39,40]. To do this, the rate of transcription for each gene was calculated based on the slope of the expression value of the labeled mRNA just prior to the peak of nascent transcript levels, while the rate of decay was determined by the slope of the unlabeled mRNA expression value following the peak of stabilization (Supplementary Data 2). The transcription and decay rates of genes with peak expression during the early ring, mid-ring, late ring/early trophozoite, mid-trophozoite, late trophozoite, and schizont stages were grouped together. The rate of transcription and decay are represented in Fig. 2d, e for each phase of development. These data show that genes expressed during the early and mid-ring stage are transcribed at a slower rate (median = 1.77 and 2.6 gene transcripts/min, respectively), compared to the rest of the life-cycle (median = 4.1–6.3 gene transcripts/min) (Fig. 2d, Supplementary Data 2). Conversely, the mean rate of decay varies significantly as the parasite progresses through the IDC (Fig. 2e; Supplementary Data 2), suggesting that the regulation of mRNA turnover is stage-specific. Surprisingly, the median rate of decay is slowest during the mid- to late-ring stages and becomes significantly more rapid during the mid-

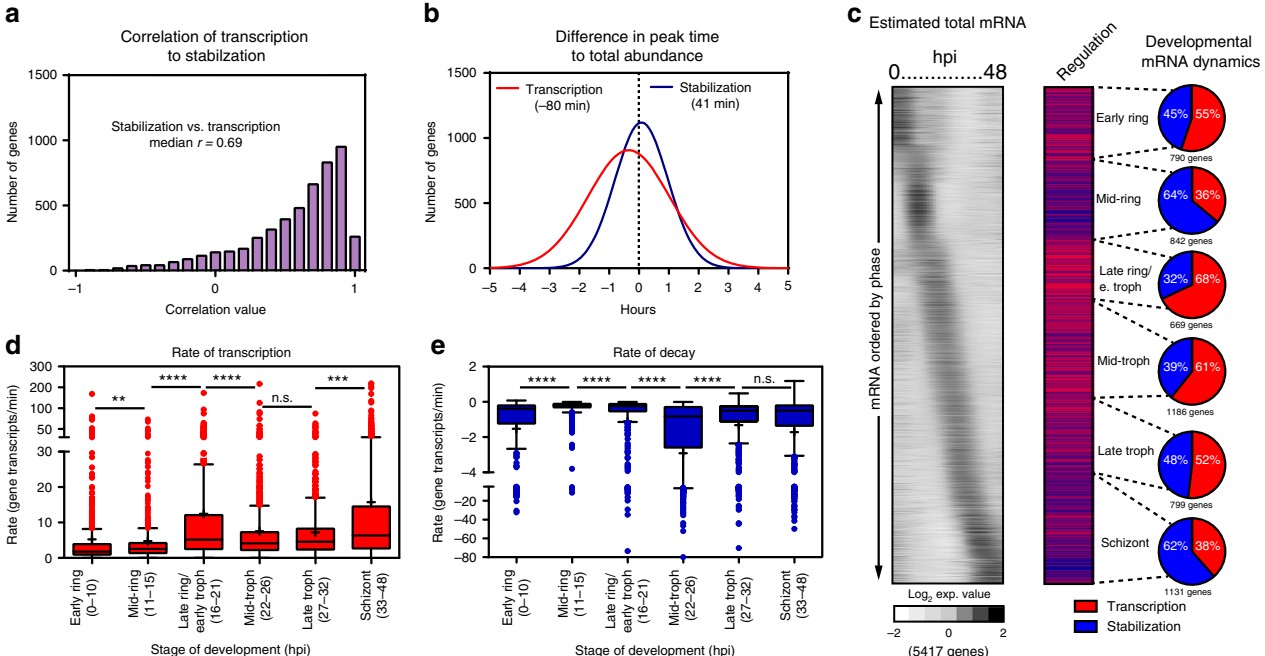

**Fig. 2** Temporal mRNA dynamics during the IDC of *P. falciparum*. **a** The correlation values of gene transcription and stabilization are summarized in a histogram plot with a median value of $r = 0.69$. **b** The difference in the peak time of nascent gene transcription (red line) and stabilization (blue line) to the peak time of mRNA abundance for each gene was calculated and displayed as a histogram. **c** The dominant mode of regulation was determined for each gene by calculating the correlation of the modeled transcription or stabilization ($r \geq 0.5$) to the estimated total mRNA abundance profile for each gene throughout the IDC and are displayed in order of the peak timing of abundance (hpi). The percentage of genes regulated by either transcription (red) or stabilization (blue) is summarized for major morphological stage transitions: early ring (790 genes), mid-ring (842 genes), late-ring/early-trophozoite (669), mid-trophozoite (1186 genes), late-trophozoite (799), and schizont stage (11,310). Box-whisker plot of **d** transcription and **e** decay rates calculated for early ring, mid-ring, late ring/early trophozoite, mid-trophozoite, late trophozoite, and schizont stages of parasite development (Supplementary Data 2). The mean rate (+) and Tukey's distribution are displayed for each stage of development. Adjusted *p* values from one-way ANOVA comparisons are noted (**$p = 0.006$; ***$p = 0.0002$, ****$p < 0.0001$). Rates that are outliers are displayed as individual points

trophozoite stage of parasite development (Fig. 2e). However, there are an increased number of genes that are stabilized during the later stages of parasite development (Fig. 2e positive outliers 36–48 hpi = 21 genes, Supplementary Data 2). Although the increase in stability of a small number of genes during the late IDC stage confirms earlier findings by Shock et al.[40], it is important to note that the vast majority of genes with peak abundance during the late trophozoite and schizont stage do not display a significantly lower rate of decay (Fig. 2e).

**IDC stage transitions coincide with "bursts" in mRNA dynamics.** Although it is conventionally accepted that transcription during the *Plasmodium* IDC occurs in a "just-in-time" fashion[10], this is largely based on transcript abundance measurements and annotated gene function. Using the measured mRNA dynamics, we examined the timing of active transcription and stabilization of the parasite as it develops throughout the IDC. By binning the genes based on their peak time of nascent transcription or peak time of stabilization every 30 min throughout the IDC, our data reveal peaks in the numbers of genes being transcribed and/or stabilized at four major time periods during parasite development (Fig. 3 and Supplementary Data 3). Interestingly, these "bursts" of mRNA regulatory activity coincide with major morphological transitions during the IDC (Fig. 3). Next, we determined the total number of genes within each "burst" by placing the boundaries at the nearest minimum trough adjacent to each peak and then performed GO-term enrichment analysis on all genes. The first peaks of both transcription and stability occurs during the mid-ring stage of development (12 hpi), and is characterized by peak transcription of genes whose products are enriched for RNA

processing, translation, and antigenic variation (Fig. 3 and Supplementary Data 3). During this stage, genes encoding for variant surface antigens and cell adhesion also reach their peak stability (Fig. 3 and Supplementary Data 3). As expected, peaks of gene regulation that occur in the early-, mid-, and late-trophozoite are enriched for genes whose functions are necessary and unique for each phase of trophozoite development (Supplementary Data 3). For example, the mid-trophozoite is known to be the most metabolically active stage of development, and during this phase, there is an enrichment in transcription and stabilization of genes, which encode for amino acid, tRNA, ncRNA, pyruvate, glycolytic, and carbohydrate metabolic processes (Fig. 3 and Supplementary Data 3). In contrast, the late-trophozoite phase is characterized by a peak in the regulation of genes involved in DNA metabolism and replication in preparation for mitotic production of daughter cells.

Beyond the identification of genes enriched during developmental transitions, we also identified a number of genes whose peak timing of transcription and stabilization occurred at either 1 or 48 hpi. As expected, at 48 hpi the number of stabilized genes (378 genes) outnumbers those that are actively transcribed (140 genes) (Fig. 3 and Supplementary Data 3). This corroborates earlier studies demonstrating the increased presence of mRNA-interacting proteins during late schizogony[27] and the increased half-life of genes at this stage[40]. As expected, these genes are enriched in processes necessary for parasite invasion and the establishment of a niche within the erythrocyte that will allow for the uptake and utilization of nutrients from the extracellular space (Supplementary Data 3), biological process GO-term enrichment, vesicle-mediated transport (GO:0006888,

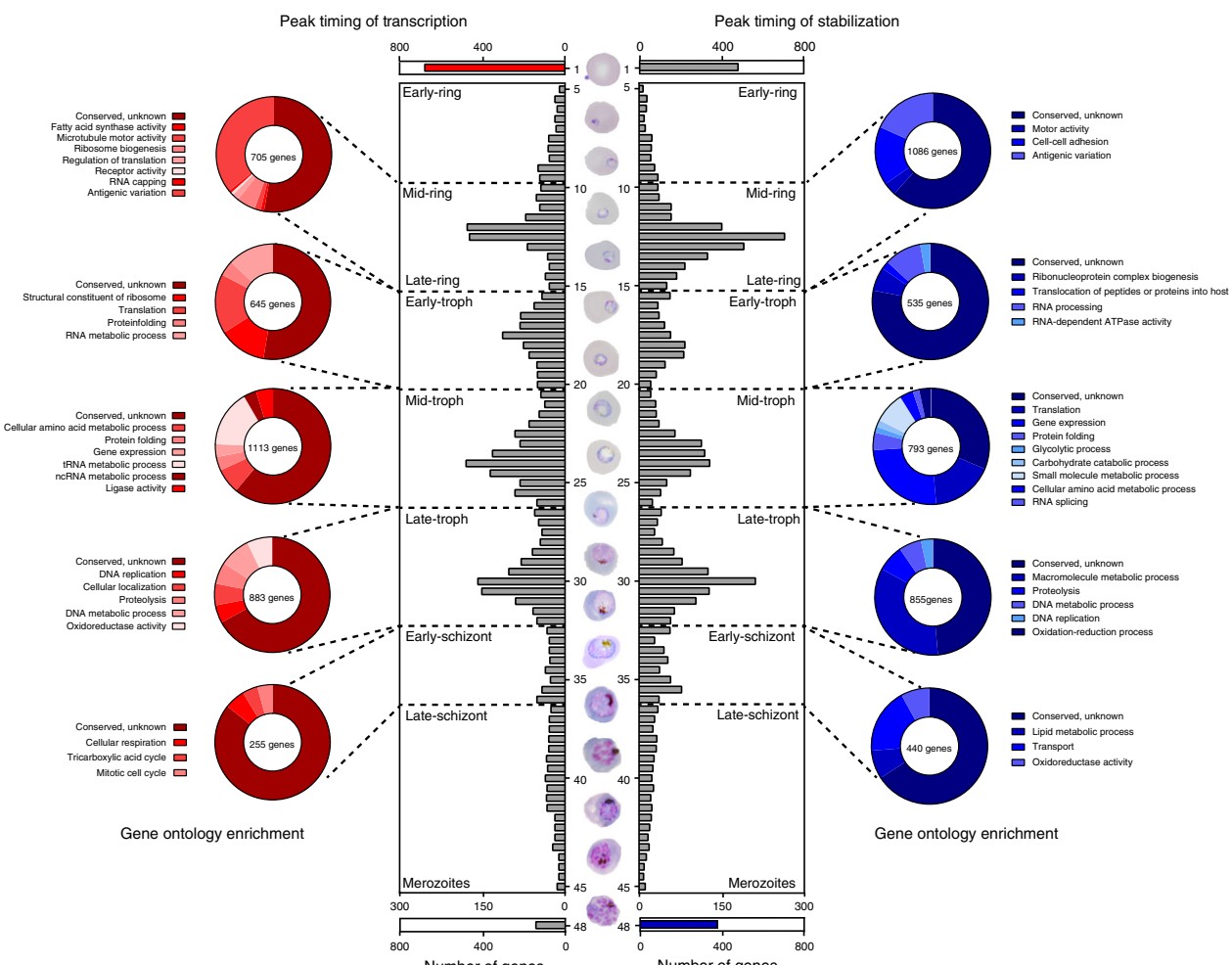

**Fig. 3** Capture of *P. falciparum* mRNA dynamics reveal "bursts" of regulation during the IDC. Histogram plot displaying the number of genes with peak transcription (left) or stabilization (right) timing binned into 30 min intervals throughout the IDC representing the hours in which there are greater than five genes enriched (5–45 hpi). The histogram is divided into five distinct groups based on the distribution of the modes and adjacent minima, corresponding with major morphological stage transitions of the IDC. Separate histograms displaying the number of genes which peak in transcription (red) and stabilization (blue) at either 1 or 48 hpi highlighting the reliance on a specific regulatory mechanism at each stage of development. The transcribed (red) or stabilized (blue) genes within each group were analyzed for GO-term enrichment and significant terms (Bonferroni corrected *p*-value ≤ 0.01) are displayed in pie charts and listed in Supplementary Data 3

Fisher's exact test, Bonferroni corrected *p*-value = 4.52e−3), and cytoskeletal protein binding (GO:0008092, Fisher's exact test, Bonferroni corrected *p*-value = 8.59e−10) including kinase signaling and microtubule formation. Conversely, genes with peak transcription occurring immediately following invasion outnumber any other hour of the IDC (Fig. 3; peak transcription 1 hpi, 684 genes; Supplementary Data 3). These genes are enriched for RNA metabolic processes (GO:0016071, Fisher's exact test, Bonferroni corrected *p*-value = 7.96e−5), likely in preparation to increase cellular transcription levels necessary for parasite development and aid in post-transcriptional regulatory processes. Identification of genes stabilized prior to or transcribed upon invasion may facilitate the dissection of the cellular and molecular processes that govern the establishment of the parasite within a new erythrocyte and provide molecular targets for future investigation.

**Nascent transcription captures clonally variant expression.** One advantage of biosynthetic mRNA capture is the ability to detect and compare the level of transcriptional activity for individual genes. Due to the importance of parasite-derived erythrocyte surface antigens in modulating immunity to malaria infections,

transcriptional regulation of these antigens is one of the most highly-studied areas of *Plasmodium* biology[60]. It has long been established that members of surface antigen gene families (*vars*, *rifins*, *stevors*, *pfmc-2tms*, and *surfins*) are transcribed during the late ring to early trophozoite stages of development and exhibit variation in their expression among parasite clones[60]. To identify the timing of transcription, we calculated the average normalized expression values (log₂ mean centered) for each gene family to determine the peak time of nascent transcription during the IDC (Fig. 4a). Transcription of the *var* and *rifin* gene families peaked at 11 hpi, while *stevors* reached their peak transcription at 12 hpi (Fig. 4a). Interestingly, the peak expression of *surfins* and *pfmc-2tm* genes occur at 10 and 16 hpi, respectively (Supplementary Fig. 6A), which may be explained by the demonstration that the transcriptional regulation of these two families occurs through different factors[61]. To determine which member(s) of these surface antigen gene families was being transcribed in this clonal population of transgenic parasites, we determined the expression values of each gene at the peak hour of transcription for each family of surface antigens (Fig. 4b). Our data demonstrate transcription of a single dominant *var*, as well as multiple *rifin* and *stevor* genes (Fig. 4b), supporting the mutually exclusive

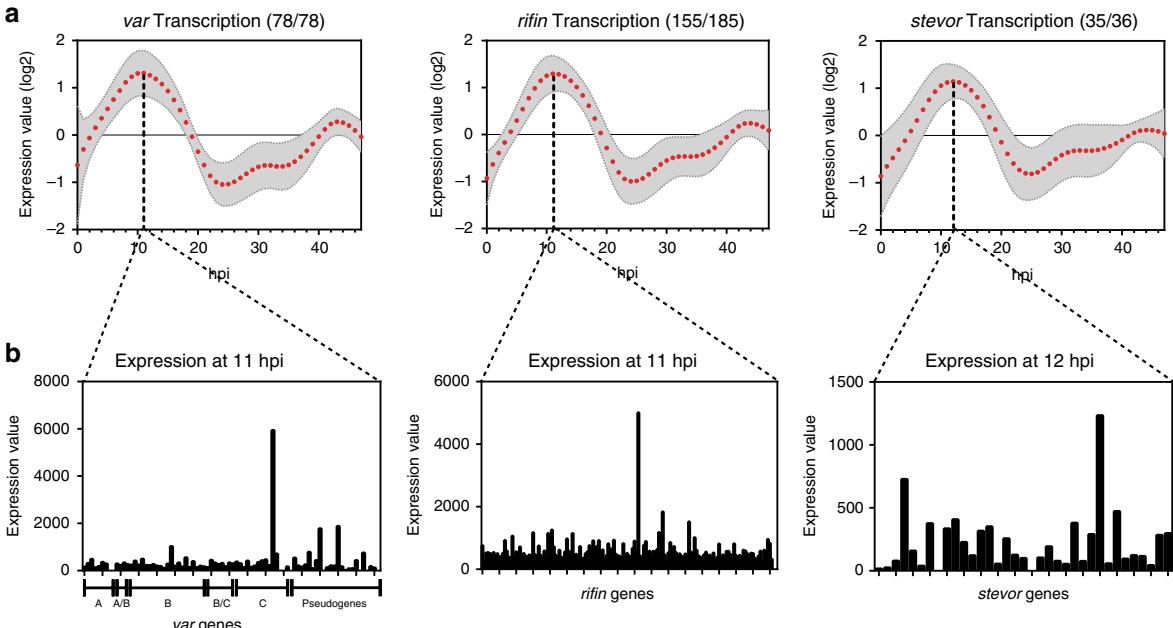

**Fig. 4** Clonal expression of *P. falciparum* variant surface antigens. **a** Transcription of surface expressed gene families: *vars* (78/78), *rifins* (155/185), and *stevors* (35/36) is displayed as the mean (log₂ expression value) and ±s.e.m. throughout the 48 h IDC. The hpi where peak transcription of each gene family is displayed as a dropline in each graph. **b** At the peak time of transcription, the expression value for each *var* (sub-groups and pseudogenes are labeled), *rifin*, and *stevor* genes

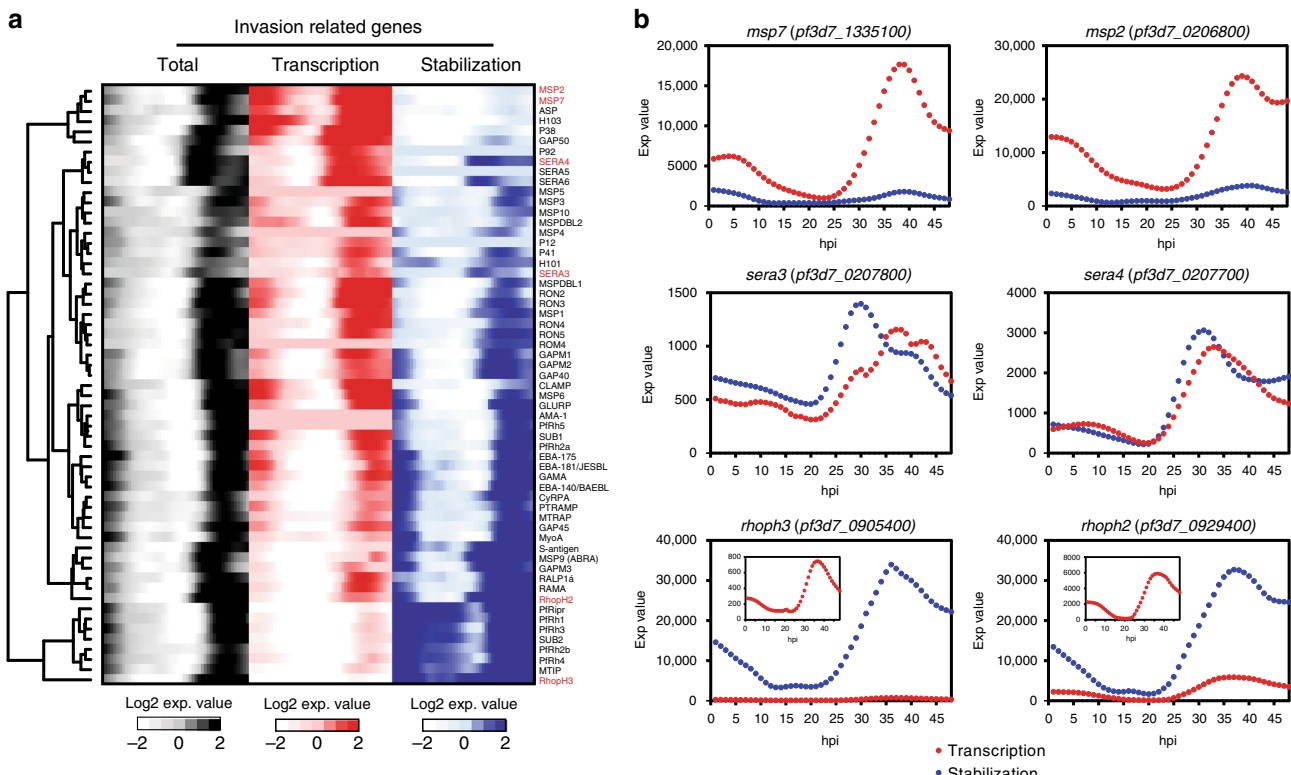

**Fig. 5** mRNA dynamics of invasion-related genes. **a** The calculated total abundance (black), transcription (red), and stabilization (blue) expression values of selected genes demonstrated to be involved in parasite erythrocyte invasion are represented and clustered based on Euclidian distance metric. Highlighted in red text are selected genes representative of the various mRNA dynamics, which regulate expression. **b** The profiles of the expression of values of transcription (red circles) and stabilization (blue circles) for selected genes representing the dominant mode of regulation through nascent transcription (*msp7* and *msp2*), a balance of both transcription and stabilization (*sera3* and *sera4*), or stabilization alone (*rhoph3* and *rhoph2*) are plotted as throughout the IDC. The insets in *rhoph2* and *rhoph3* plots are the expression values of transcription from 0 to 48 hpi

expression of only the *var* gene family of surface antigens. Although *pfmc-2tm* and *surfin* genes are also considered surface antigens, they do not demonstrate clonal variation (Supplementary Fig. 5B), as reported previously[62]. Taken together, the biosynthetic labeling and capture of nascent transcription detects transcriptional variation within clonally variant gene families.

**mRNA regulation of invasions genes is predictive of function.** A closer examination of the mRNAs stabilized during the schizont stage of development revealed that a majority encode proteins that function in the invasion of merozoites into erythrocytes. Because of this observation, we determined if these mRNAs are co-regulated by comparing their similarity in total abundance, transcription, and stabilization (Fig. 5a). As previously demonstrated[10], we found that the total mRNA abundance profiles of invasion genes peak during schizogony (Fig. 5a). However, the levels of active transcription and stabilization vary drastically (Fig. 5a, b), revealing that there are at least two distinct regulatory mechanisms that dictate these patterns of total abundance. For example, a majority of mRNAs that encode merozoite surface proteins (*msps*), including *msp7* and *msp2*, are predominantly regulated at the level of transcription and appear to be minimally influenced by stabilization (Fig. 5a, b). This may suggest that *msps* are rapidly transcribed and translated during schizogony for presentation on the surface of the merozoite prior to rupture from the red blood cell[63]. Conversely, although rhoptry genes *rhoph2* and *rhoph3* are also transcribed during the schizont stage, their peak abundance is contributed by mRNA stabilization (Fig. 5b), suggesting that these mRNAs may be translationally

repressed or are slow to turn over until their gene products are necessary. Previous reports have demonstrated a "step-wise" role for proteins involved in invasion, which is required after the merozoite has formed a tight interaction with the erythrocyte surface[64]. More recently, invasion proteins (such as RhopH2 and RhopH3) have been shown to assist in the establishment of nutrient transporters across the parasitophorous vacuole membrane following invasion[63,65–67]. Although these genes are transcribed during the schizont stage and their protein products are necessary for invasion, stabilization guarantees that the mRNAs will be available for translation immediately upon invasion to promote proper nutrient transport for the developing parasite as recently demonstrated[65,66]. Interestingly, the mRNA dynamics of both *sera3* and *sera4* reveal a balance of transcription and stabilization (Fig. 6b), potentially due to their dual role in two stages of the IDC, rupture as well as reinvasion. The *SERA* proteins have been demonstrated to be essential for schizont rupture and remain associated with the merozoite surface until attachment and invasion of a new red blood cell[68].

**mRNA dynamics are predictive of *cis*-regulatory sequence motifs.** High resolution capture of mRNA dynamics provides a more refined set of data for the identification of DNA regulatory motifs associated with co-transcription. Using the finding informative regulatory elements (FIRE) algorithm[69], we used co-regulated mRNAs to predict DNA sequence motifs that are enriched in the 5′ untranslated regions of these genes (up to −1000 bp from ATG start site) (Fig. 1c and Supplementary Data 1). We found enrichment of 21 DNA motifs associated with

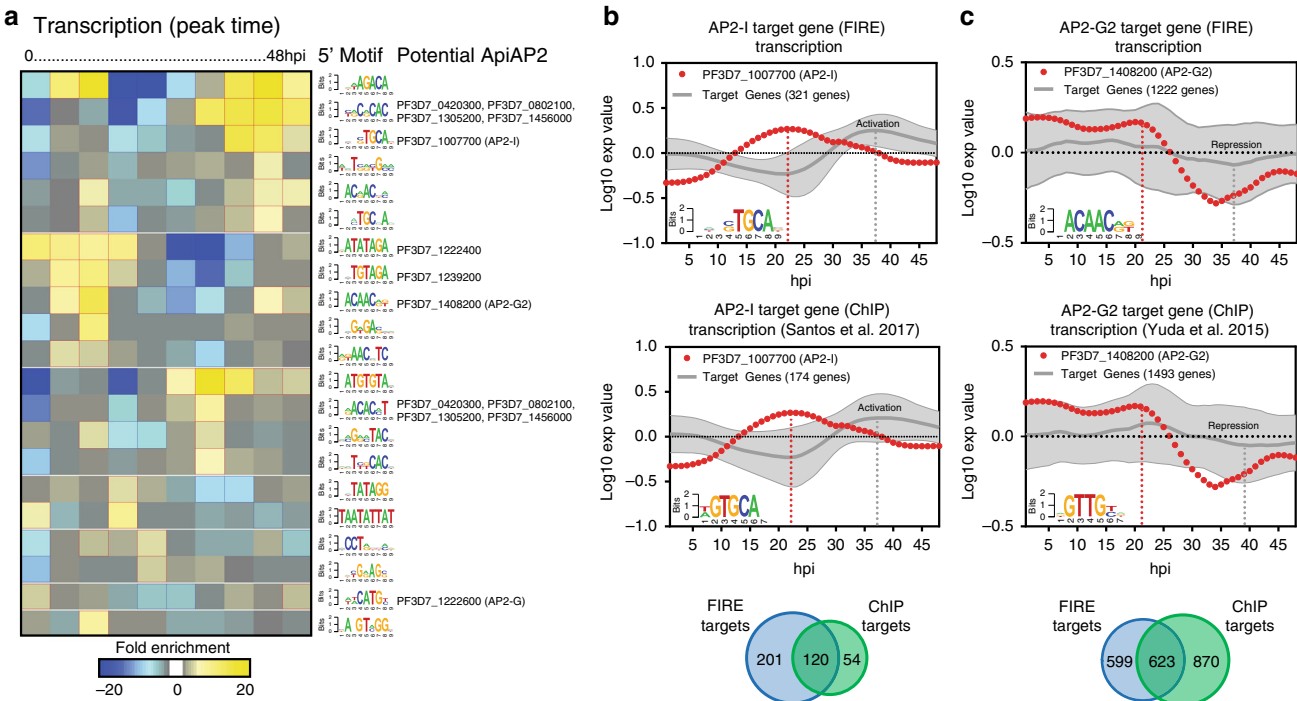

**Fig. 6** Prediction of *cis*-regulatory motifs from *P. falciparum* mRNA dynamics. **a** FIRE analysis prediction of enriched DNA and RNA motifs within the whole genome dynamics data. The profiles of transcription and stabilization are clustered into peak timing during the IDC (columns) predicted enriched motifs arranged into putative functional modules (rows). The yellow color map indicates overrepresentation of a motif in a given cluster; significant overrepresentation (*p* < 0.05 after Bonferroni correction) is highlighted with red frames. Similarly, the blue color map and blue frames indicate underrepresentation. For each motif, the ApiAP2 *trans*-factor demonstrated to interact with each predicted motif[70] is listed. **b** The nascent transcription profile of genes identified by FIRE (left panel) or ChIP-seq (right panel[71]) with the enriched *Pf*AP2-I motif GTGCA are plotted (gray, mean log[2] expression value and ±s.e.m.) in comparison to the transcription profile of *pfap2-i* (red circle). Venn Diagram represents the overlap of genes identified by FIRE (blue) or ChIP-seq (green). **c** The nascent transcription profile of genes identified by FIRE (left panel) or ChIP-seq (right panel[72]) with the enriched *Pf*AP2-G2 motif ACAAC are plotted (gray, mean log[2] expression value and ± s.e.m.) in comparison to the transcription profile of *pfap2-g2* (red). Venn Diagram represents the overlap of genes identified by FIRE (blue) or ChIP-seq (green)

co-transcription during the IDC (Fig. 6a). Of these 21 motifs, seven have been previously identified as ApiAP2 cis-interacting motifs[70]. To verify the accuracy of our motif identification, we calculated the average transcription of genes enriched for the known DNA recognition motif (GTGCA), which is bound by the third AP2 domain of PfAP2-I[70], compared to the expression of pfap2-i (Fig. 6b). As expected for a transcriptional activator, genes containing the GTGCA regulatory motif are actively transcribed following the peak expression of pfap2-i (Fig. 6b). Additionally, our bioinformatic analysis identified 120 genes previously demonstrated to be directly regulated by PfAP2-I via ChIP-seq analysis (Fig. 6c)[71]. Conversely, transcription of genes enriched for the cis-regulatory motif ACAAC decreases following the transcription of ap2-g2 (Fig. 6c), which has been suggested to act as a transcriptional repressor during the IDC in Plasmodium berghei[72]. We compared the target genes enriched with the ACAAC DNA-motif with the AP2-G2 ChIP-seq gene targets and found that capture of nascent transcription, combined with FIRE analysis, identified greater than 50% of the reported targets of AP2-G2[72] (623/1222) (Fig. 6c). Taken together, using motif identification analysis, we can predict the potential regulatory role of ApiAP2 proteins based upon the transcriptional patterns of genes containing their cognate DNA-interaction motif compared to the mRNA dynamics of individual ApiAP2s (Supplementary Fig. 7A). Finally, we used the nascent transcription data to predict the possible regulatory roles for ApiAP2s proteins for which no identified cis-motif has been reported[70]. To do this, we compared the timing and intensity of all genes enriched for the orphan AGACA motif (Fig. 6a) to the transcriptional pattern of two functionally uncharacterized ApiAP2 protein-coding genes: pf3D7_1139300 and pf3D7_0404100. The peak timing for both of these apiap2 genes occurs just prior to a decrease in transcription of genes enriched with the AGACA motif (Supplementary Fig. 7B) (as seen for AP2-G2 (Fig. 6c)).

## Discussion

Cellular RNA levels are subject to extensive regulation involving alterations in the rates of RNA synthesis (transcription), processing (e.g., splicing, polyadenylation, transport), stabilization, and decay[35]. Data on gene expression levels in P. falciparum obtained by microarray or RNA-seq[7,10,11,36,62] reflect steady-state quantities of intracellular RNA, which are a product of the rate of both genome-wide transcription and stabilization. However, a major constraint of these analyses is the inability of these methods to differentiate between transcription and stabilization, which contribute to global abundance profiles throughout the IDC. As we have demonstrated, this constraint can be overcome by 4-TU biosynthetic labeling of newly transcribed mRNA providing direct isolation and analysis of nascent transcripts with minimal perturbations and toxic effects to the cell[25].

Here, we genetically modified P. falciparum to constitutively express the pyrimidine salvage pathway by single-copy genome integration, enabling the rapid incorporation of 4-TU into nascent mRNA transcripts. Using these parasites, we labeled and captured nascent transcription and mRNA stability and determined their contribution to the total abundance profiles of each gene throughout intraerythrocytic development. Earlier transcriptome studies captured timing and mRNA abundance of each gene by DNA microarray analysis[5,10], revealing a periodicity to gene expression during the 48 h IDC. Here, we demonstrate that the periodicity of each gene is defined by contributions from both transcription and stabilization. However, mRNA regulation is also time-dependent and gene-specific. Given the complexities of genome-wide mRNA dynamics, we examined the contribution of mRNA regulation to gene expression as the parasite progresses through the IDC revealing that transcription and stabilization do not contribute equally to the patterns of abundance observed for a given gene. Our results identify active transcription throughout the IDC (Fig. 1c), in contrast to previous reports, which suggest that there is minimal transcriptional activity during the ring and schizont stages of development[26]. We also find that there are distinct periods of transcriptional and post-transcriptional regulatory bursts, which coincide with breaks in protein production at the late-ring and trophozoite stages of development[21], supporting the suggestion that replenishment of proteins at these stages is regulated at the level of gene expression[19]. Our data also support previous studies, which suggest that the half-life of the transcripts increases during the intraerythrocytic developmental cycle[40]. However, this slow rate of decay applies to less than 25 genes and the majority of genes undergo a fairly consistent rate of decay through the IDC. Therefore, the degradation machinery is consistently active throughout the IDC[28] and it is more likely that post-transcriptional regulation leads to stabilization of this small subset of genes.

Beyond the examination of genome-wide P. falciparum mRNA dynamics throughout the IDC, these global transcriptional dynamics provide gene-specific profiles of nascent transcription and stabilization. From this, we distinguished regulatory dynamics of genes with similar patterns of abundance and determine mutually exclusive expression of single genes within clonally variant families. Co-transcription and co-stabilization of genes provide a useful resource for the detection of cis-acting regulatory motifs and, the prediction of motifs with associated trans-factors. These orphan motifs can now be used to identify new DNA-binding proteins, which remain uncharacterized. Our data provides a comprehensive resource that will enable further investigation into the gene regulatory systems and mRNA metabolism of the malaria parasite.

Future studies will need to clarify the mechanisms regulating genome-wide rates of decay by evaluating the RNA-interacting proteins that regulate transcript stability throughout the IDC. By combining biosynthetic mRNA labeling and capture with cross-linking and mass spectrometry[73–75], a complete RNA–protein interactome of developmentally regulated transcripts can now be achieved. Additional future applications of this method will also focus on measuring immediate transcriptional responses to environmental or chemotherapeutic treatments. This data set not only provides the malaria research community with measurements of the RNA dynamics for all individual genes throughout the IDC, but will also serve to enhance the bioinformatics analysis of transcription models, co-expression networks, and will aid in defining the timing of biological processes.

## Methods

**Transgene construction.** The open reading frame of the yeast FCU gene was PCR amplified from P. falciparum vector pCC1[76], and cloned as a translational fusion into the unique AvrII and BsiWI sites of the pLN-ENR-GFP vector[52], resulting in pLN-5′cam-FCU-GFP plasmid and placing transcriptional control under the P. falciparum calmodulin promoter (pf3d7_1434200). This plasmid was transformed into, replicated in, and isolated from DH5α Escherichia coli for transfection into P. falciparum 3D7attBattb.

**Strains and culture maintenance.** P. falciparum strain 3D7attB was obtained from BEI Resources (cat # MRA-845) as a mycoplasma-free frozen stock and has been described in previous studies[52]. Parasite cultures were maintained under standard conditions[77] at 5% hematocrit of O+ human erythrocytes (Biological Specialty Corporation) in RPMI1640 containing hypoxanthine, NaH2CO3, HEPES, glutamine and 5 g/L AlbuMAX II (Life Technologies). The 3D7attB line has been genetically altered to contain an acceptor attB site (from Mycobacterium smegmatis) into the non-essential cg6 locus. This recombinant locus is maintained by continuous selection with 2.5 nM WR99210 (Jacobus Pharmaceuticals, Princeton, NJ)[78].

**Generation of genetically modified parasite lines.** Transfection of P. falciparum strain 3D7 (BEI Resources, Cat # MRA-102) was performed as previously described[78]. Briefly, 5–7% ring-stage parasite cultures were washed three times with

10 times the pellet volume of cytomix. The parasitized RBC pellet was resuspended to 50% hematocrit in cytomix. In preparation for transfection, 100 μg of both pLN-5′cam-FCU-GFP and pINT plasmid, expressing a *Mycobacterium* Bxb1 integrase that mediates *attP* and *attB* recombination, were precipitated and resuspended in 100 μL cytomix. The plasmid and 250 μL of the 50% parasitized RBC suspension were combined and transferred to a 0.2 cm electroporation cuvette on ice. Electroporation was carried out using a BioRad GenePulser set at 0.31 kV, 960 μF. The electroporated cells were immediately transferred to a T-25 flask containing 0.2 mL uninfected 50% RBCs and 7 mL medium. To select for parasites containing both plasmids, medium containing 1.5 μg/μL Blasticidin S (Sigma-Aldrich) and 250 μg/mL G418 Sulfate (Geneticin®, Thermo Fisher Scientific) was added at 48 h post transfection. Cultures were maintained under constant 1.5 μg/μL Blasticidin S (Sigma-Aldrich) pressure, splitting weekly, until viable parasites were observed. Viable transgenic parasites were then cloned by limiting dilution and genomic DNA isolated to detect the presence of the integrated fusion gene of *fcu-gfp* was PCR verified using previously published primer pairs[52].

**Verification of transgene expression.** Western blot analysis of FCU-GFP protein expression was carried out on mixed stage transgenic *P. falciparum* (10% parasitemia, 5% hematocrit), isolated by saponin (0.01%) lysis. Protein extracts from both 3D7[attB] wild-type parasites and 3D7[attB::FCU-GFP] (clone A2) transgenic parasites were run on a SDS-10% polyacrylamide gel and transferred to a nitrocellulose membrane. The membrane was blocked with 5% milk for 30 min. To detect protein, the membrane was exposed to sheep IgG anti-cytosine deaminase primary antibody (Thermo Fisher Scientific, cat # PA1–85365) diluted 1:500 in 3% bovine serum albumin (BSA) and incubated overnight at 4 °C followed by a 1 h incubation with donkey anti-sheep HRP secondary antibody (Thermo Fisher Scientific, cat# A16041) diluted 1:1000 in 3% BSA. The membrane was incubated for 1 min in Pierce® ECL-reagent (Thermo Fisher Scientific) and protein detected by exposure to film.

**Verification of in vivo FCU-GFP protein expression by IFA.** FCU-GFP expression in transgenic parasite lines was verified by immunofluorescence imaging analysis. Mixed stage transgenic *P. falciparum*-infected erythrocytes were fixed with 4% formaldehyde/0.0075% glutaraldehyde (Polysciences, Inc.) using the method of Tonkin et al.[79]. Fixed cells were permeabilized with 0.01% Triton-X 100 (Sigma-Aldrich) and blocked with 10% normal goat serum and 3% BSA. To detect FCU-GFP localization, the suspension was incubated with a 1:1000 dilution of a mouse anti-GFP antibody (Thermo Fisher Scientific) followed by secondary antibody Alexaflour-488 rat anti-mouse (Molecular Probes) diluted to 1:500. Cells were resuspended in Slowfade Antifade reagent with DAPI to stain parasite nuclei (Thermo Fisher Scientific) and mounted on slides using Flouromount-GTM (Southern Biotech). Fluorescent microscopic images were obtained with an Olympus BX61 system and deconvoluted using SlideBook 5.0 software (Intelligent Imaging Innovations).

**Assessment of FCU-GFP salvage of 4-TU via Northern blot.** Function of the FCU-GFP transgene was verified in the 3D7[attB::fcu-gfp] strain compared to wild-type 3D7[attb] by the addition of 40 μM of 4-TU (Sigma Aldrich) to the culture medium from a 200 mM stock concentration prepared in DMSO as previously described[25]. After parasites were grown in the presence of 4-TU for 12 h, total RNA was prepared from parasites in 5 mL of TRIzol and resuspended in DEPC-treated water to a final concentration of 0.5 μg/μL. As described in Painter et al.[25], total RNA was biotinylated with EZ-link Biotin-HPDP dissolved in formamide to a concentration of (Thermo Fisher Scientific) for 3 h at room temperature under RNase-free conditions to reduce the amount of RNA degradation that can occur. Biotinylated RNA (2.5 μg) was run on an Ethidium Bromide 1% agarose gel at 4 °C for 30 min at 200 V and was transferred to Hybond-N+ nitrocellulose (Amersham) membrane using traditional northern blotting techniques. The RNA was UV-crosslinked to the membrane and probed with streptavidin–HRP (1:1000) (Thermo Fisher Scientific). Biotinylated RNA was detected via incubation with ECL-reagent and exposure to film.

**Thiol-labeling timecourse.** *P. falciparum* transgenic parasites were cultured in O+ human erythrocytes (Biological Specialty Corporation) at 5% hematocrit in RPMI1640 complete medium as described above. All transgenic parasites were synchronized with three successive (48 h apart) L-alanine treatments to remove late stage parasites[80]. To ensure a sufficient amount of mRNA was labeled and captured for further analysis, we grew 48 individual parasite cultures to 15% parasitemia in 100 mL of medium at 5% hematocrit, one culture flask (150 cm²) for each time-point. Upon invasion, 40 μM 4-TU was added to the medium and parasites were incubated for 10 min followed by total RNA isolation with TRIzol. 4-TU addition, incubation, and RNA extraction was repeated every hour for 48 successive time-points throughout intraerythrocytic development.

**Biosynthetic modification and isolation of labeled mRNA.** Briefly, transgenic *P. falciparum* was incubated with 4-TU and total RNA was extracted as described above. All RNA was subjected to the biosynthetic modification and magnetic separation as previously published with a few adjustments[44,46]. Specifically, 80 μg of total RNA (at a concentration of 0.4 μg/μL) was incubated at room temperature protected from light for 3 h in the presence of 160 μL of 1 mg/mL solution of EZ-link Biotin-HPDP (Thermo Fisher Scientific). Biotinylated total RNA was

precipitated and resuspended in DEPC-treated water to a final concentration of 0.5 μg/mL. Incorporation of 4-TU and biotinylation was determined by NanoDrop analysis and Northern blot probed with streptavidin–HRP. 4-TU labeled-biotinylated RNA was purified using Dynabeads® MyOne™ Streptavidin C1 magnetic beads (Life Technologies) at a concentration of 2 μL/μg of RNA. Beads were pre-washed as per manufacturers protocol to remove any RNases and blocked with 16 μg of yeast tRNA (Life Technologies). 4-TU labeled-biotinylated RNA was added to the bead slurry and incubated at room temperature for 20 min with rotation. The RNA-bead slurry was placed on a magnetic stand for 1 min, and liquid carefully removed and saved for RNA precipitation. This sample contained RNA that was not thiolated or biotinylated. Then the RNA-bound beads underwent five rounds of stringent washes with buffer consisting of 1 M NaCl, 5 mM Tris–HCl (pH 7.5), 500 μM EDTA in DEPC-treated water. 4-TU labeled-biotinylated RNA was eluted from the beads with 5% 2-mercaptoethanol incubated for 10 min at room temperature, placed back on the magnetic stand, liquid removed and saved for RNA precipitation. The RNA in this fraction contained mRNA transcribed in the presence of 4-TU from FCU-GFP expressing cells.

**Reverse transcription and Agilent DNA microarray analysis.** Total RNA, unbound, and bound fractions of mRNA are precipitated by traditional methods (0.1 vol NaCl, 0.1 vol linear acrylamide and 3× volume 100% EtOH) followed by resuspension in 20 μL DEPC-treated water. The concentration of each sample is determined by NanoDrop ND-1000. Starting with 1.0–2.5 μg of RNA, single-strand aminoallyl-containing cDNA synthesis and Amersham CyDye-coupling (GE Healthcare) was carried out as previously described[10] with the addition of control RNAs to aid in the normalization between samples and arrays (two color RNA spike-in kit, Agilent Technologies). To prevent Cy5 degradation by ozone, all steps starting with dye resuspension were carried out in an ozone-free environment. Final cDNA concentration and dye-incorporation was assessed on a NanoDrop ND-1000 spectrophotometer. To reduce sample and time-point bias, cDNA from each time-point was labeled with Cy5 and combined with an equal amount of Cy3-labeled cDNA reference pool generated from equal amounts of ring, trophozoite, and schizont stage mRNA and hybridized to *P. falciparum* custom arrays (Agilent Technologies 60mer SurePrint platform, AMADID #037237). Hybridized arrays were incubated for 16 h in a rotating hybridization oven (10 rpm) at 65 °C. Prior to scanning, arrays were washed in 6× and 0.06× SSPE (both containing 0.005% N-lauryl-sarcosine) (Sigma-Aldrich, St. Louis, MO, USA), followed by an acetonitrile rinse.

**Array scanning and data acquisition.** Arrays were scanned on an Agilent G2505B Microarray Scanner (Agilent Technologies) with 5 μm resolution at wavelengths of 532 (Cy3) and 633 nm (Cy5) using the extended dynamic range (10–100%) setting. Normalized intensities were extracted using Agilent Feature Extraction Software Version 9.5 employing the GE2_1100_Jul11_no_spikein extraction protocol.

**Array data normalization and expression value calculation.** Each set of 48 arrays for total, labeled, and unlabeled RNA was normalized using the Rnits (v1.2.0) from the Bioconductor project in R[53]. Arrays for 45 hpi contained an abundance of extreme outliers and were excluded from further analysis. Prior to data smoothing and fitting, probes that are Agilent Controls, rRNAs (28 probes), or found to be not unique (133 probes) were removed. The timecourse profiles of each probe were then smoothed using cubic splines and all negative values set to 0.5. For each probe, contribution of unlabeled (stabilized) and labeled (transcribed) RNAs was modeled to sum up to the total abundances, assuming that contributions are naturally non-negative:

$$\text{Total mRNA abundance} = \alpha(\text{nascent transcription}) + \beta(\text{mRNA stabilization})$$

In this equation, $\alpha$ and $\beta$ are the coefficients applied to the probe intensity values on the labeled and unlabeled RNA arrays, respectively. This problem of non-negative least squares is solved by the Lawson–Hanson algorithm (the nnls package in R). As a simple verification that the expression values estimated for both transcription and stabilization were accurate, the predicted total mRNA abundance was computed by adding the estimated unlabeled and labeled expression values. The hourly pairwise-Pearson correlation between the predicted total mRNA abundances and the conventional observed abundances was calculated between every timepoint. From this, the median Pearson correlation was determined from the directly corresponding hourly timepoints. Each probe was summarized by gene using Tukey's biweight with a parameter $c = 5$. Finally, three timecourses for each gene are plotted over 48 h. These data are available on http://www.PlasmoDB.org[56] on individual gene pages or for download of the full dataset.

**Calculating peak timing and rates of transcription.** We utilized the Lomb-Scargle periodogram approach to determine the peak time of total abundance, transcription, and stabilization for each gene as described previously[81] using the available LombScargle package for R. The resulting peak times for each gene can be found in Supplementary Data 1. These peak times were used to order genes based on peak time of total abundance and to bin genes based on their peak times of transcription and stabilization.

Rates of transcription for each gene were determined based upon the slope of the calculated expression values from the start of transcription to the adjacent peak. Conversely, rates of decay were calculated by determining the slope of the expression values of unlabeled RNA (stabilized fraction) from the peak to the following trough. Calculated rates of transcription and decay for each gene were then binned into groups based upon the peak timing of nascent transcription or stabilization of that gene (Supplementary Data 2). These groups represent six major morphological stage transitions; early ring (0–10 hpi), mid-ring (11–15 hpi), late ring/early trophozoite (16–21 hpi), mid-trophozoite (22–26 hpi), late trophozoite (27–32 hpi), and schizont (33–48 hpi). The calculated rates are plotted as a box-whisker plot with Tukey's Biweight distribution and median represented.

**Motif and GO-term enrichment analysis**. We identified motifs enriched in the 5′ UTRs of genes co-transcribed throughout the IDC using a regulatory element discovery algorithm (FIRE)[69] and these results are represented as a heatmap. The activity profile of each motif was used to identify a refined list of target genes that were both enriched for the motif and similarly transcribed. Gene lists identified by FIRE analysis to be enriched for either the AP2-I or AP2-G2 motif were compared to the previously published target gene listed identified by ChIP-seq[71,72].

GO-term enrichment of select gene groups or clusters was carried out using the Analysis Tool at http://www.PlasmoDB.org[56] and terms that are significantly enriched (Fisher's exact test, Bonferroni corrected $p$-value ≤ 0.05) are summarized in Supplementary Data 3.

**Data availability**. DNA microarray data that was generated for this study are available on Gene Expression Omnibus (https://www.ncbi.nlm.nih.gov/geo) under study accession number GSE66669 and on figshare with the identifier https://doi.org/10.6084/m9.figshare.6200792.v1. The authors declare that all other data supporting the findings of this study are available within the article and its Supplementary Information files, or are available from the authors upon request.

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

## Acknowledgments

This work was funded through support from NIH R21 AI133379. A.S. is supported through the J. Lloyd and Dorothy Foehr Huck Institutes of the Life Sciences at Penn State University. We would like to thank many people who have contributed to this project including Simon Cobbold, Bjorn F.C. Kafsack, Jess O'Hara, Lindsey Orchard, Yoanna Pompalova, Joana Santos, April Williams, and the Genomics Core Facility at Princeton University and its staff, especially Donna Storton and Jessica Buckles Wiggins. We also thank Omar Harb, Brian Brunk, and Shon Cade at http://www.PlasmoDB.org for making this data publicly available ahead of publication. Lastly, we thank Erik Allman, Gabrielle Josling, Scott Lindner, and Timothy Russell for critical reading of the manuscript.

## Author contributions

H.J.P. and M.L. designed the experiments. H.J.P. performed experiments. H.J.P. and N.C.C. analyzed the data. H.J.P. and M.L. wrote the manuscript and generated figures. N.C.C. and J.D.S. created and implemented the statistical model used for data analysis. A.S. and I.A. contributed to data analysis and public availability for hosting the data. M.L. and J.D.S. provided critical comments and project oversight.

## Additional information

**Competing interests:** The authors declare no competing interests.

