## [Peer Review File · Nature Communications]

Reviewers' comments:

Reviewer #1 (Remarks to the Author):

The dynamic changes in the *P. falciparum* transcriptome have been described well at the level of transcript abundance, but in this manuscript, Painter et al. show that to understand the underlying mechanisms, we really must look separately at the rates of mRNA synthesis and decay, since both processes are independently regulated in Plasmodium, as in other eukaryotes. Here, the authors exploit a genetic approach they published recently and used to study the transcriptome during sexual commitment, to measure mRNA half-lives during the intraerythrocytic developmental cycle (IDC) of *P. falciparum* asexual blood stages. They rule out strong effects of gene length and dUTP content on label incorporation and develop a statistical model to assess the relative contribution of transcription and stabilisation of existing mRNA molecules to the total amount of transcript for each gene and time point. There is clear evidence from this paper that both, transcription and decay are independently regulated and contribute to transcript abundance, leading to the conclusion there are stage and gene specific changes in turn-over of mRNAs.

The data presented will be an important resource for the Plasmodium research community and will allow future genome-scale analyses to take into account the dynamic aspects of gene expression. It is commendable that the data have already been released through the community database, PlasmoDB. In addition to functioning as a resource, the data reveal some intriguing new aspects of Plasmodium biology.

The sharp bursts of regulation shown in Figure 3 at 30 min resolution are particularly astonishing. Parasite cultures for this experiment have gone through three rounds of alanine synchronisation, so I guess they may be synchronised to a 3-5 h window. How can Fig. 3 show sharp bursts at ~ 1 h resolution? Would peak transcript abundance reveal a similar, possibly weaker pattern? If not, why not? If yes, is that also present in the previously published high resolution transcriptomic time courses, such as the one in the initial Bozdech et al. microarray study? Are there environmental or treatment factors that could have acted at the level of individual culture flasks to produce the observed patterns? Given that this was an hourly time course over 48 hours, which was presumably conducted only once, were there factors with a ~ 6 h periodicity that could produce the observed pattern, such as shift changes in the lab, temperature fluctuations during labelling due to new batches of labelling medium or batch effects from processing six samples together? If Fig. 3 reveals a hidden artefact in the data, how would that affect the dataset as a whole?

If the regulatory bursts are real, they would presumably be consistent with the discrete transcriptional states observed by Reid et al. in single cell data (eLife, 2018). Could the authors do a comparison of the genes involved and consider this in their discussion? It would be fascinating to develop a single cell read-out for 4-TU incorporation, although outside the scope of this study.

On P16 the authors suggest RhopH2 and H3 transcripts are stabilized until after invasion to modify the PV. How would that be consistent with these proteins being present in rhoptries,

which no longer exist after invasion and with there being very little total mRNA from these genes in rings (Fig. 5A)? It seems striking that there are two different patterns of transcription/stabilization regulating of invasion related genes, both achieving the same overall profile of mRNA abundance. Instead of assigning different functions to these, I would argue that under some circumstances it may be of limited functional consequence through which combination of transcription and stabilization a given mRNA abundance profile is produced. In support of this, I would point to the observation that proteins known to interact, such as RhopH2 and RhopH3, members of the motor complex and of the Rh5 complex, are split across different parts of the dendrogram in Fig. 5A. Perhaps combining two types of regulation with translational repression serves to give the parasite a wider window during which to produce large quantities of the most abundant transcripts for the late stage proteins?

Other points:

Introduction: "27-member Apicomplexan AP2 (ApiAP2) proteins" should be "protein family"?

P7: dUTPs/per gene

P11: Sentence starting "These data show that genes expressed during...". Why is 1.77 transcript/min faster than 4.1?

Figures S1: The legend needs attention. The first mention of D) seems to be a second reference to B). The legend to what seems to be the actual panel D) (northern blot?) contains a reference to a bottom panel, which does not exist. It would also be nice to have an indication of marker mobility on the blots to see if the bands have the expected size.

Fig. 2B, upper right-hand panel. It seems wrong from the raw data shown that normalisation reduces the stabilisation curve to zero at all times points. Has this been plotted correctly?

Fig. 2D, E. Differences are referred to as significant in the text but not statistics are provided to back this up.

Reviewer #2 (Remarks to the Author):

This is a clearly written manuscript that describes an extensive survey of gene expression for an important human pathogen enabled by genetic engineering, microarray technology and sophisticated analysis. The data are very accessible and provide reasonable support for the authors' claims

This work describes the creation of a genetically modified Pf strain that facilitates pulse-labeling to flag newly-transcribed RNA. The nascent mRNA transcripts were separated by

affinity capture, fluorescence labeled, and hybridized to Pf DNA microarray. In Parallel, the remainder fraction of mRNA (termed "stabilized") as well as an aliquot not subjected to fractionation (termed "total RNA") was assayed as well.

By pulsing for 10 minutes followed by RNA isolation each hour of the 48 hr development cycle (IDC), authors were able to follow full transcriptome mRNA expression over this time course and simultaneously estimate the contribution of new transcript production in vivo at any particular time point for any gene on the microarray.

The authors performed extensive analysis of the set of microarray data (48 time points x 3 RNA samples x 5,198 genes) including mathematical modeling and normalization of each gene over time and with respect to the three mRNA estimates (nascent fraction, remainder fraction, total), and gene function annotation analysis to interpret the rate of transcription as it pertains to the roles of gene families and individual genes in particular stages of the life cycle of the malaria parasite.

Major Issues:

A key concept in the analysis is that a given gene will have a life-cycle stage specificity in its expression. This is described in previous microarray work on the Pf 3D7 strain (reference 13) the same treatment of the data produces a plot that appears very similar but the demonstration of agreement – the matrix of all pairwise correlations (Figure S2B) shows only approximate agreement.

The data would be significantly improved by inclusion of replicate microarrays, preferably from independent time course trials. The presented design has no replication.

The comparison of the total (which includes potential perturbations due to the 4-TU labeling), to previous work (reference 13) is deemed "no significant changes in pattern" yet the plot of pairwise correlations – appears to reveal a non-linear correspondence of the time courses. I assume "same pattern" would have a clear trend of maximal positive correlation along the diagonal in Figure S2B. It is unclear how exactly a "median corr = 0.72" was calculated, what the corr statistic is, or what value of "corr" would indicate a significant change.

Figure S2D, another analysis which would demonstrate a lack of bias due to thiol capture, is hard to evaluate without kernel density contours.

There seems to be a problem with the data provided in table S1. I merged the data for Total Transcribed, Stabilized, Estimated Total and examined the correlation between the "Peak Time" values. I assumed that the Peak Time based on the Total RNA, should be well correlated with the Peak Time based on the Estimated Total RNA. But it looks like no

correlation exists (See attached "peak time compare I.png"). As expected, there is a strong correlation between Peak Time from Estimated Total and the other two (Transcribed and Stabilized). And it is interesting to see the discordance between time points associated with Transcribed vs. Stabilized (attached "peak time compare II.png").

Lack of internal consistency and lack of agreement with previously measured time course suggests that the further interpretations of the dynamics may be faulty.

Minor:

For describing this study, I would argue that "global" is a poor replacement for "genome-wide" especially in context of investigating malaria pathogen, a disease of important epidemiological interest, where it is confusing at first take. I understand that "global dynamics" has been coined in other nature communications papers, but still.

Figure S2A: Traditionally, in descriptions of microarray experiments – "labeling" refers to the fluorescent labeling for detection. Suggest to use "biotinylated" instead of "labeled" misplaced : "non biotinylated"

Hybridized "against" (how about "pooled reference (cy3)")

"newly transcribed" instead of "Transcribed" ?

Control tests for possible interaction of biotinylation with indirect (amino allyl) fluorescent labeling were apparently not done.

Page 6: "(4-TU labeled)" should be "(4-TU labeled, biotinylated)"

Page 7 "median corr = 0.72" was this this a Pearson, Spearman?

Page 25: "Feature Extractor software" should be "Feature Extraction software"

figure 3 legend: "p-value \geq 0.01" was probably supposed to be "p-value $<$ 0.01" ?

Reviewer #3 (Remarks to the Author):

The manuscript by Painter et al. describes the use of a relatively unperturbed biological system to acquire detailed measurements of active transcription and transcript stabilization throughout the asexual replication cycle of the human malaria parasite *P. falciparum*. While the data is novel, it is mostly presented as a resource and lacks significant new biological insight. My two main concerns about this manuscript are:

1) The study is built on the premise that transcription occurs in a "just-in-time" fashion, resulting in a transcriptional gradient over the course of the cell cycle. The data obtained in

this study reinforce this convention and show a good correlation with microarray data obtained years ago. However, a limitation of this and earlier studies is that the work is done on bulk populations of parasites. While the parasite cultures were synchronized prior to the collection of RNA, the synchronization that is achieved for *P. falciparum* cultures is far from perfect. An inherent problem of working with bulk populations is therefore that there is always a level of asynchrony. A recently published study using single-cell RNA-seq did not replicate the transcriptional gradient across the cell cycle, but instead found evidence for a transcriptional switch and a clear delineation between the transcriptional programs of trophozoite- and schizont-stage parasites (Reid et al, eLIFE 2018). While I understand that the study by Painter et al. requires bulk parasite populations, I am asking the authors to discuss the limitations of their study and to present the alternative view of switching transcriptional programs during the asexual cycle in addition to the “conventionally accepted” view of a transcriptional gradient.

2) The authors use a 10-minute 4-TU labeling pulse to distinguish between transcripts synthesized within this 10-minute window and transcripts that were already present prior to the addition of 4-TU. The latter population is termed “stabilized” transcripts. On page 6, the authors included a short explanation that the “stabilized” transcripts are merely unlabeled and not necessarily bound by an RBP or subject to a low turnover rate. Since these transcripts are not “stabilized” per se, I find this term confusing and would recommend to change it. In this respect, I also wonder how much value the analysis of “mRNA stabilization” has, since the unlabeled transcripts seem to be mostly a product of the fact that genes are transcribed for longer periods of time (i.e. longer than 10 minutes) and transcripts are maintained for more than 10 minutes in the cell to ensure sufficient time for processing, transport and translation. In Figure 3, I am therefore not surprised that there is a large overlap in GO terms associated with transcription/stabilization among each group. The term stabilization should be reserved for transcripts with a significant delay between peak of nascent transcription and peak abundance in the unlabeled fraction, and I would be interested in seeing an additional analysis of this subgroup of transcripts.

Minor comments:

Typo in Fig. S4A: “Orangellar” should read “Organellar”

Reviewers' comments:

Reviewer #1 (Remarks to the Author):

The dynamic changes in the *P. falciparum* transcriptome have been described well at the level of transcript abundance, but in this manuscript, Painter et al. show that to understand the underlying mechanisms, we really must look separately at the rates of mRNA synthesis and decay, since both processes are independently regulated in *Plasmodium*, as in other eukaryotes. Here, the authors exploit a genetic approach they published recently and used to study the transcriptome during sexual commitment, to measure mRNA half-lives during the intraerythrocytic developmental cycle (IDC) of *P. falciparum* asexual blood stages. They rule out strong effects of gene length and dUTP content on label incorporation and develop a statistical model to assess the relative contribution of transcription and stabilisation of existing mRNA molecules to the total amount of transcript for each gene and time point. There is clear evidence from this paper that both, transcription and decay are independently regulated and contribute to transcript abundance, leading to the conclusion there are stage and gene specific changes in turn-over of mRNAs.

The data presented will be an important resource for the *Plasmodium* research community and will allow future genome-scale analyses to take into account the dynamic aspects of gene expression. It is commendable that the data have already been released through the community database, PlasmoDB. In addition to functioning as a resource, the data reveal some intriguing new aspects of *Plasmodium* biology.

The sharp bursts of regulation shown in Figure 3 at 30 min resolution are particularly astonishing. Parasite cultures for this experiment have gone through three rounds of alanine synchronisation, so I guess they may be synchronised to a 3-5 h window. How can Fig. 3 show sharp bursts at ~1 h resolution? Would peak transcript abundance reveal a similar, possibly weaker pattern? If not, why not? If yes, is that also present in the previously published high resolution transcriptomic time courses, such as the one in the initial Bozdech et al. microarray study? Are there environmental or treatment factors that could have acted at the level of individual culture flasks to produce the observed patterns? Given that this was an hourly time course over 48 hours, which was presumably conducted only once, were there factors with a ~6h periodicity that could produce the observed pattern, such as shift changes in the lab, temperature fluctuations during labelling due to new batches of labelling medium or batch effects from processing six samples together? If Fig. 3 reveals a hidden artefact in the data, how would that affect the dataset as a whole?

As interpreted by the reviewer, the asexual parasite cultures used in this study are tightly synchronized within a 3-5 hour window. With respect to Fig. 3 and our calculation of the peak times of transcription and stabilization for each gene, we will elaborate on how the data was generated and processed for the resulting histogram: First, the data generated in this study are representative of the patterns of gene transcription and stabilization from a population of parasites and are, therefore, interpreted in the context of genome-wide expression over time. We agree that our window of synchrony is beyond the set bin time; however, the expression level of each gene has been normalized using both time and multiple probes in comparison with data across an entire genome. From this we are able to generate a smoothed "expression profile" for each gene and determine the "Peak Time" (maximum) along that curve which is representative of the behavior of each gene within a population of parasites as shown below (Response Fig. 1). To determine the maximal nascent transcription and stabilization for each gene, we calculated the "Peak Time" during the asexual lifecycle using Lomb-Scargle analysis. Figure 3 in the manuscript represents the number of genes whose peak transcription or stability occurs throughout the 48 hour lifecycle within arbitrary 30 minute bins. These calculated "Peak Times" of abundance, transcription, and stabilization for each gene and corresponding p-values are supplied in Table S1, columns XY and XZ.

Figure 1: Expression Values representing nascent transcription (red), stabilization (blue), and total mRNA abundance of *pf3d7_071600* over 48 hours of asexual development. The peak times for each represented profile are noted by dotted lines.

We agree that the “bursts” of gene expression (and regulation) are surprising and believe that this is due to the high resolution, percent genome coverage, and culture synchronicity of this study. Interestingly, we do see this same pattern in our total abundance timecourse (Response Fig. 2A). As requested, if we compare these “bursts” to the peak times of abundance from previously published 48 hour mRNA timecourses from Bozdech *et al* 2003 [1] and Llinás *et al.* 2006 [2], these same patterns are apparent, although they are less distinct (Response Fig. 2B and 2C). It is likely that the lower genome coverage of these earlier DNA microarray studies and the reduced culture synchrony contribute to the decreased detection of this pattern in these earlier studies. Because this pattern is not unique to our current study, we do not believe that it is a result of batch effects in sample collection or processing. In addition, extreme care was taken with experimental design, sample preparation, and data normalization to reduce or remove any batch effects that could influence the results of the study. For example, shift changes in sample collection varied from 4 to 11 hours (only one shift change consisted of 6 hours), sample processing was performed in 8 hour batches, and DNA microarray hybridization was carried out in 24 sample batches. All data was first normalized within a timecourse, followed by inter-timecourse calculations of non-negative least squares to determine the relative contribution of transcription and stabilization to total abundance.

Figure 2: Histogram of calculated peak time of total abundance for each gene throughout the 48h asexual development cycle. A) Comparison of peak time of total abundance versus estimated total abundance in *P.f.* 3D7attb::FCU-GFP. Histogram representation of peak time of mRNA abundance for each gene in prior full 48 hour IDC DNA microarray studies from *P.f.* strains B) 3D7 and C) HB3.

If the regulatory bursts are real, they would presumably be consistent with the discrete transcriptional states observed by Reid et al. in single cell data (eLife, 2018). Could the authors do a comparison of the genes involved and consider this in their discussion? It would be fascinating to develop a single cell read-out for 4-TU incorporation, although outside the scope of this study.

We thank this reviewer and reviewer #3 for suggesting a comparison of our data to recent scRNA-seq data. We direct reviewer #1 to our response to reviewer #3 below for a more thorough comparison of the genes described in discrete transcriptional states by Reid *et al.* 2018 [3]. We agree that it would be fascinating to develop a single cell read-out for out nascent transcription labeling method. Interestingly, as we show below, using our biosynthetic mRNA labeling method we are able to detect the same general transcript abundance patterns from a population of cells as those seen via scRNA-seq.

On P16 the authors suggest RhopH2 and H3 transcripts are stabilized until after invasion to modify the PV. How would that be consistent with these proteins being present in rhoptries, which no longer exist after invasion and with there being very little total mRNA from these genes in rings (Fig. 5A)? It seems striking that there are two different patterns of transcription/stabilization regulating of invasion related genes, both achieving the same overall profile of mRNA abundance. Instead of assigning different functions to these, I would argue that under some circumstances it may be of limited functional consequence through which combination of transcription and stabilization a given mRNA abundance profile is produced. In support of this, I would point to the observation that proteins known to interact, such as RhopH2 and RhopH3, members of the motor complex and of the Rh5 complex, are split across different parts of the dendrogram in Fig. 5A. Perhaps combining two types of regulation with translational repression serves to give the parasite a wider window during which to produce large quantities of the most abundant transcripts for the late stage proteins?

As you accurately point out, RhopH2 and H3 proteins are present in the rhoptries, which no longer exist following invasion. However, two recent studies have suggested that RhopH2 and H3 also play a role in the establishment of the parasitophorous vacuole. We argue that it is possible that the stability of the transcripts which encode for RhopH2 and H3 are due to the essentiality of both of these gene products in establishing the parasitophorous vacuole and nutrient uptake [4, 5]. We agree that it is likely that RhopH2 and H3 are regulated by both transcription and the more dominant mechanism of stabilization/translational repression. The text has been edited to clarify this point in the manuscript as follows:

“Although these genes are transcribed during the schizont stage and their protein products are necessary for invasion, stabilization guarantees that the mRNAs will be available for translation immediately upon invasion to promote proper nutrient transport for the developing parasite as recently demonstrated^{91,92}.”

Other points:

Introduction: “27-member Apicomplexan AP2 (ApiAP2) proteins” should be “protein family”?

The text has been edited per the reviewer’s suggestions.

P7: dUTPs/per gene

The text has been edited per the reviewer’s suggestions.

P11: Sentence starting “These data show that genes expressed during...”. Why is 1.77 transcript/min faster than 4.1?

This error has been corrected in the main text.

Figures S1: The legend needs attention. The first mention of D) seems to be a second reference to B). The legend to what seems to be the actual panel D) (northern blot?) contains a reference to a bottom panel, which does not exist. It would also be nice to have an indication of marker mobility on the blots to see if the bands have the expected size.

We thank the reviewer for directing our attention to the errors in the figure legend for Fig. S1. The legend has been corrected to reflect proper description of the panels.

Fig. 2B, upper right-hand panel. It seems wrong from the raw data shown that normalisation reduces the stabilisation curve to zero at all times points. Has this been plotted correctly?

Yes, the data represented in the upper panel of Fig. 2B has been plotted correctly. In this case, following normalization, the contribution of mRNA stabilization to the total abundance profile is negligible. Please refer to Table S1 for the data represented in this plot.

Fig. 2D, E. Differences are referred to as significant in the text but not statistics are provided to back this up.

Figure 2D has been updated to reflect the significance associated with these differences in decay rates throughout the IDC as described in the text.

Reviewer #2 (Remarks to the Author):

This is a clearly written manuscript that describes an extensive survey of gene expression for an important human pathogen enabled by genetic engineering, microarray technology and sophisticated analysis. The data are very accessible and provide reasonable support for the authors' claims.

This work describes the creation of a genetically modified Pf strain that facilitates pulse-labeling to flag newly-transcribed RNA. The nascent mRNA transcripts were separated by affinity capture, fluorescence labeled, and hybridized to Pf DNA microarray. In Parallel, the remainder fraction of mRNA (termed "stabilized") as well as an aliquot not subjected to fractionation (termed "total RNA") was assayed as well.

By pulsing for 10 minutes followed by RNA isolation each hour of the 48 hr development cycle (IDC), authors were able to follow full transcriptome mRNA expression over this time course and simultaneously estimate the contribution of new transcript production in vivo at any particular time point for any gene on the microarray.

The authors performed extensive analysis of the set of microarray data (48 time points x 3 RNA samples x 5,198 genes) including mathematical modeling and normalization of each gene over time and with respect to the three mRNA estimates (nascent fraction, remainder fraction, total), and gene function annotation analysis to interpret the rate of transcription as it pertains to the roles of gene families and individual genes in particular stages of the life cycle of the malaria parasite.

Major Issues:

A key concept in the analysis is that a given gene will have a life-cycle stage specificity in its expression. This is described in previous microarray work on the Pf 3D7 strain (reference 13) the same treatment of the data produces a plot that appears very similar but the demonstration of agreement – the matrix of all pairwise correlations (Figure S2B) shows only approximate agreement.

The data would be significantly improved by inclusion of replicate microarrays, preferably from independent time course trials. The presented design has no replication.

The comparison of the total (which includes potential perturbations due to the 4-TU labeling), to previous work (reference 13) is deemed “no significant changes in pattern” yet the plot of pairwise correlations – appears to reveal a non-linear correspondence of the time courses. I assume “same pattern” would have a clear trend of maximal positive correlation along the diagonal in Figure S2B. It is unclear how exactly a “median corr = 0.72” was calculated, what the corr statistic is, or what value of “corr” would indicate a significant change.

While we agree that any study could be improved by inclusion of biological replicates, this is beyond the scope of the study. Furthermore, time series experiments conducted at high resolution (hourly timepoints in this case) are internally controlled. As you know, only a handful of 48 hour transcriptome studies with hourly resolution have been performed over the past 15 years[1, 2], all of which are internally controlled and demonstrate a high overall concordance between datasets and transcriptomes from various strains. We also note that due to differences in synchronization, the Pearson correlation in published DNA microarray and RNA-sequencing timecourses studies looking at various strains of *P. falciparum* range from 0.70 to 0.85 [2, 6]. To further demonstrate that our data has a similar correlation to previously published data sets, we have calculated the hourly pair-wise Pearson correlation statistic between 3D7attb::FCU-GFP total abundance (this study) and two other strains of *P. falciparum*, HB3 and DD2, from Llinás *et al.* 2006 [2] (Response Fig. 3). The median Pearson correlation value was calculated from corresponding hourly timepoints between 3D7attb::FCU-GFP and HB3 (median corr = 0.77) or DD2 (median corr = 0.801). We have updated our methods to clarify this calculation as follows:

“The hourly pairwise-Pearson correlation between the predicted total mRNA abundances and the conventional observed abundances was calculated between every timepoint. From this, the median Pearson correlation was determined from the directly corresponding hourly timepoints.”

Figure 3: Pearson correlation heatmap comparing 48 hourly timepoints between *P. falciparum* strains 3D7attb::FCU-GFP (from this study) and HB3 or DD2 (from Llinas *et al.* 2006).

Figure S2D, another analysis which would demonstrate a lack of bias due to thiol capture, is hard to evaluate without kernel density contours.

We agree that this is a good suggestion, and we have edited Figure S2D to include 2D kernel density contours as shown below:

There seems to be a problem with the data provided in table S1. I merged the data for Total Transcribed, Stabilized, Estimated Total and examined the correlation between the "Peak Time" values. I assumed that the Peak Time based on the Total RNA, should be well correlated with the Peak Time based on the Estimated Total RNA. But it looks like no correlation exists (See attached "peak time compare I.png"). As expected, there is a strong correlation between Peak Time from Estimated Total and the other two (Transcribed and Stabilized). And it is interesting to see the discordance between time points associated with Transcribed vs. Stabilized (attached "peak time compare II.png").

We heartily thank the reviewer for thoroughly inspecting the data provided and acknowledge that Table S1 provided the incorrect peak timing for Total RNA. This resulted from an excel column sorting issue during preparation of Table S1 and fortunately does not reflect an error in the analysis, nor does it impact the data interpretation or display. The correct peak times are now supplied in a properly sorted version of Table S1 and we would be happy to provide the reviewer with the appropriate output files of the Lomb-Scargle analysis by which the peak timing was calculated to ensure clarity of our process. To further demonstrate the agreement between samples, we performed a correlation analysis between the peak times for each gene in each sample as the reviewer had done (Response Fig. 4)

Figure 4: Correlation dot plot comparing peak time of total mRNA abundance for each gene in *P.f.* 3D7attb::FCU-GFP versus the estimated total abundance. Histograms represent the number of genes with peak times calculated at 30 min intervals throughout the 48 hour asexual development cycle.

Lack of internal consistency and lack of agreement with previously measured time course suggests that the further interpretations of the dynamics may be faulty.

We hope that our above replies to the reviewers are sufficient to quell their concerns regarding the interpretations of mRNA dynamics during the IDC of *P. falciparum*. Again, we apologize for this oversight.

Minor:

For describing this study, I would argue that “global” is a poor replacement for “genome-wide” especially in context of investigating malaria pathogen, a disease of important epidemiological interest, where it is confusing at first take. I understand that “global dynamics” has been coined in other nature communications papers, but still.

To reduce any confusion that the term “Global” may introduce, we have altered our title as follows:

“ Genome-wide Real-time *in vivo* Transcriptional Dynamics During *Plasmodium falciparum* Blood-Stage Development”

Figure S2A: Traditionally, in descriptions of microarray experiments – “labeling” refers to the fluorescent labeling for detection. Suggest to use “biotinylated” instead of “labeled” – We have altered “Labeled” to “Thiol-labeled” misplaced : “non biotinylated” – This accurately describes that the total RNA pool (containing thiol-modified mRNA) was not biotinylated

Hybridized “against” (how about “pooled reference (cy3)”) – We have altered our description of the “Pooled RNA Reference” to “Pooled RNA Reference (Cy3)”

“newly transcribed” instead of “Transcribed”? – “Transcribed” has been altered to read “Newly Transcribed”

While we agree with the reviewer that conventional descriptions of microarray experiments include “labeling” in reference to fluorescent dye incorporation, in this study the “labeling” specifically refers to thiol-incorporation. Since these descriptors were used in our prior publication of this method [7], we prefer to keep the terminology consistent.

Control tests for possible interaction of biotinylation with indirect (amino allyl) fluorescent labeling were apparently not done.

Perhaps an elaboration of the methodology will clear up the confusion regarding the different roles of biotinylation (for affinity purification of 4-thiol uracil labeled RNAs versus Cy dye incorporation into aminoallyl-dUTP-containing cDNA generated from these RNAs. We do not see a need for determining any possible interaction of biotin with the amino allyl label because the thiol-group incorporated into nascent mRNA is biotinylated with EZ-link Biotin-HPDP (Thermo Scientific) that conjugates via a cleavable (reversible) disulfide bond allowing for the removal of the modified mRNA away from the biotin, which remains bound to streptavidin magnetic beads. The eluted mRNA is then used as a template to reverse transcribe aminoallyl-dUTP cDNA for subsequent Cy dye-coupling and microarray hybridization. The likelihood of free biotin or biotinylated mRNA remaining after reverse transcription is very small; however, to address this, two rounds of DNA cleanup and column purification were used throughout.

Page 6: “(4-TU labeled)” should be “(4-TU labeled, biotinylated)”

The text has been edited per the reviewer’s suggestions.

Page 7 “median corr = 0.72” was this this a Pearson, Spearman?

The correlation statistic is a Pearson correlation. The text has been edited to clarify.

Page 25: “Feature Extractor software” should be “Feature Extraction software”

The text has been edited per the reviewer's suggestions.

figure 3 legend: "p-value ≥ 0.01 " was probably supposed to be "p-value < 0.01 " ?

Yes. Thank you for catching our error. The text has been edited per the reviewer's suggestions.

Reviewer #3 (Remarks to the Author):

The manuscript by Painter et al. describes the use of a relatively unperturbed biological system to acquire detailed measurements of active transcription and transcript stabilization throughout the asexual replication cycle of the human malaria parasite *P. falciparum*. While the data is novel, it is mostly presented as a resource and lacks significant new biological insight. My two main concerns about this manuscript are:

We thank the reviewer for acknowledging that the method utilized and data captured in this study are a novel and powerful resource. As suggested, the data presented in this study will hopefully be a useful resource to the malaria community and provide genome-wide transcriptional and post-transcriptional activity throughout the 48h asexual development cycle. However, we respectfully disagree with the reviewer that this study lacks biological insight and, although the manuscript has been submitted as a Resource Article, we have presented novel findings with respect to gene-regulatory transitions (Fig. 3), calculated rates of transcription and stabilization (Fig. 2), and variation in methods of regulation of genes involved in a single biological process (Fig. 5) during asexual development. Below we have responded to the reviewer's main concerns.

1) The study is built on the premise that transcription occurs in a "just-in-time" fashion, resulting in a transcriptional gradient over the course of the cell cycle. The data obtained in this study reinforce this convention and show a good correlation with microarray data obtained years ago. However, a limitation of this and earlier studies is that the work is done on bulk populations of parasites. While the parasite cultures were synchronized prior to the collection of RNA, the synchronization that is achieved for *P. falciparum* cultures is far from perfect. An inherent problem of working with bulk populations is therefore that there is always a level of asynchrony. A recently published study using single-cell RNA-seq did not replicate the transcriptional gradient across the cell cycle, but instead found evidence for a transcriptional switch and a clear delineation between the transcriptional programs of trophozoite- and schizont-stage parasites (Reid et al, eLIFE 2018). While I understand that the study by Painter et al. requires bulk parasite populations, I am asking the authors to discuss the limitations of their study and to present the alternative view of switching transcriptional programs during the asexual cycle in addition to the "conventionally accepted" view of a transcriptional gradient.

While we agree with the reviewer that there are limitations to capturing the accurate transcriptional activation when working with bulk cultures, we disagree with the premise that our method is a mere "reinforcement" of the conventional "just-in-time" transcription and not capable of accurately distinguishing transcriptional "switches" as proposed in a more recent study [3]. As proof that our method is capable of detecting a similar transcription patterns, we first examined the genes that were identified by Reid *et al.* [3] to be involved in the *P. berghei* transcriptional switch (Response Fig. 5; top panel). A heatmap representation of the same genes shown by Reid *et al.* from the total abundance, transcribed, and stabilized datasets presented in our manuscript clearly demonstrates the same pattern of gene expression "switching" at points of morphological differentiation during asexual development. Our data also reveal a similar pattern with genes identified from Reid *et al.* in the *P. falciparum* transcriptional switch (Response Fig 5; bottom panel).

Transcriptional “Switches”

Figure 5: Heatmap representation of genes identified to represent a "transcriptional switch" in Reid *et al.* 2018 from both *P. berghei* and *P. falciparum*. The data represented are Expression Values calculated from this study which have been log2 transformed and median centered.

While we are able to capture a similar transcriptional switch as that shown by Reid *et al.* between Clusters 1/2 and Cluster 3, we believe that this selective display of the data in Reid *et al.* fails to represent the full scope of mRNA abundance during development because the genes shown are roughly 350 of the total 2,249 genes which have peak abundances and transcription during the trophozoite and schizont stages from our data. In addition, because single-cell studies are limited by the amount of RNA within a parasite, the stage of parasite development in which transcript abundance can be reliably measured is restricted to trophozoite and schizont stage parasites. Therefore, we do not agree that our “bulk” analysis is limited in its ability to detect the “switching” of transcriptional programs nor should it be considered a lesser method in the capture of transcript abundance during asexual parasite development. In fact, the data from both our study and Reid *et al.* 2018 serve to further validate an established set of genes first shown 15 years ago by Bozdech *et al.*, 2003 Figure 5 (“Phaseogram of Putative Vaccine Targets”) [1] which demonstrate sharp transcriptional activation prior to the schizont stages.

2) The authors use a 10-minute 4-TU labeling pulse to distinguish between transcripts synthesized within this 10-minute window and transcripts that were already present prior to the addition of 4-TU. The latter population is termed “stabilized” transcripts. On page 6, the authors included a short explanation that the “stabilized” transcripts are merely unlabeled and not necessarily bound by an RBP or subject to a low turnover rate. Since these transcripts are not “stabilized” per se, I find this term confusing and would recommend to change it. In this respect, I also wonder how much value the analysis of “mRNA stabilization” has, since the unlabeled transcripts seem to be mostly a product of the fact that genes are transcribed for longer periods of time (i.e. longer than 10 minutes) and transcripts are maintained for more than 10 minutes in the cell to ensure sufficient time for processing, transport and translation. In Figure 3, I am therefore not surprised that there is a large overlap in GO terms associated with transcription/stabilization among each group. The term stabilization should be reserved for transcripts with a significant delay between peak of nascent transcription and peak abundance in the unlabeled fraction, and I would be interested in seeing an additional analysis of this subgroup of transcripts.

While we do note that our method cannot distinguish whether these “stabilized” RNAs are free or bound by RBPs and are likely turning over at variable rates, multiple studies have demonstrated that the turnover rate of mRNAs in eukaryotes is greater than 10 min [8-10]. Additionally, from our data, the rate of transcription of a majority of genes (>75%) occurs in less than one min (≥ 1 transcript/min)(Fig. 2D) which is comparable to the rates of mRNA transcription, maturation and processing for a number of other organisms [8, 11-17]. Therefore, we assert that transcripts which persist for longer than 10 minutes are post-transcriptionally regulated/translationally repressed which reflects their “stabilization”. We use this term consistently throughout our manuscript and, in the absence of a term which could more accurately describe these transcripts, prefer to utilize this descriptor.

As per the reviewer’s suggestion, we have closely examined transcripts with a significant delay between the peak of nascent transcription and peak stabilization. There are 678 transcript whose peak stabilization is greater than five hours after their peak transcription, suggesting that these transcripts are post-transcriptionally regulated. Within this group of stabilized transcripts, there are only two GO-terms which are significantly enriched (GO:0006412: translation and GO:0016192: vesicle-mediated transport, p-value < 0.005) and no identifiable enriched RNA-motif which could possibly mediate interaction with a RNA-interacting protein. While we do not believe it would be impactful to add this analysis to the manuscript, we do provide the peak timing of abundance, transcription, and stabilization for each gene in Table S1 as a resource for further analysis.

Minor comments:

Typo in Fig. S4A: “Orangellar” should read “Organellar”
We have corrected this typographical error.

References:

1. Bozdech, Z., et al., *The transcriptome of the intraerythrocytic developmental cycle of Plasmodium falciparum*. PLoS biology, 2003. **1**(1): p. E5.
2. Llinas, M., et al., *Comparative whole genome transcriptome analysis of three Plasmodium falciparum strains*. Nucleic Acids Res, 2006. **34**(4): p. 1166-73.
3. Reid, A.J., et al., *Single-cell RNA-seq reveals hidden transcriptional variation in malaria parasites*. Elife, 2018. **7**.
4. Sherling, E.S., et al., *The Plasmodium falciparum rhoptry protein RhopH3 plays essential roles in host cell invasion and nutrient uptake*. Elife, 2017. **6**.
5. Counihan, N.A., et al., *Plasmodium falciparum parasites deploy RhopH2 into the host erythrocyte to obtain nutrients, grow and replicate*. Elife, 2017. **6**.
6. Otto, T.D., et al., *New insights into the blood-stage transcriptome of Plasmodium falciparum using RNA-Seq*. Molecular Microbiology, 2010. **76**(1): p. 12-24.

7. Painter, H.J., et al., *Whole-genome synthesis and decay of biosynthetically labeled transcripts during intraerythrocytic development of P. falciparum*. In preparation, 2017.
8. Rabani, M., et al., *Metabolic labeling of RNA uncovers principles of RNA production and degradation dynamics in mammalian cells*. Nature biotechnology, 2011. **29**(5): p. 436-42.
9. Peccarelli, M. and B.W. Kebaara, *Measurement of mRNA decay rates in Saccharomyces cerevisiae using rpb1-1 strains*. J Vis Exp, 2014(94).
10. Wada, T. and A. Becskei, *Impact of Methods on the Measurement of mRNA Turnover*. Int J Mol Sci, 2017. **18**(12).
11. Windhager, L., et al., *Ultrashort and progressive 4sU-tagging reveals key characteristics of RNA processing at nucleotide resolution*. Genome Res, 2012. **22**(10): p. 2031-42.
12. Barrass, J.D., et al., *Transcriptome-wide RNA processing kinetics revealed using extremely short 4tU labeling*. Genome Biol, 2015. **16**: p. 282.
13. Neymotin, B., R. Athanasiadou, and D. Gresham, *Determination of in vivo RNA kinetics using RATE-seq*. RNA, 2014. **20**(10): p. 1645-52.
14. Gray, J.M., et al., *SnapShot-Seq: a method for extracting genome-wide, in vivo mRNA dynamics from a single total RNA sample*. PLoS One, 2014. **9**(2): p. e89673.
15. Danko, C.G., et al., *Signaling pathways differentially affect RNA polymerase II initiation, pausing, and elongation rate in cells*. Mol Cell, 2013. **50**(2): p. 212-22.
16. Darzacq, X., et al., *In vivo dynamics of RNA polymerase II transcription*. Nat Struct Mol Biol, 2007. **14**(9): p. 796-806.
17. Boireau, S., et al., *The transcriptional cycle of HIV-1 in real-time and live cells*. J Cell Biol, 2007. **179**(2): p. 291-304.

REVIEWERS' COMMENTS:

Reviewer #1 (Remarks to the Author):

I am satisfied with the authors' responses to the points raised by reviewers.

Reviewer #2 (Remarks to the Author):

I believe the authors have adequately addressed the reviewers' concerns. The manuscript is improved and deserves publication in Nature Communications.